# UnifiedVerifier: Unifying Paradigms in Automated LLM Evaluation

## Abstract

The current landscape of Large Language Model (LLM) evaluation is fragmented, with bespoke models for objective verification (e.g., answer&process-verify, fact-checking) and subjective judgment (e.g., response quality ranking) operating in isolation. These models are often trained under specific task paradigms or fixed prompts, lacking versatility and failing to accommodate user needs for customizable evaluation criteria, input forms, and output formats. To address these challenges, this paper introduces UnifiedVerifier, an innovative framework designed to achieve comprehensive, general-purpose, and customizable verification capabilities within a single model. The core contributions of UnifiedVerifier are twofold: first, we present Evolutionary Verification Data Synthesis (Evo-Verify), a multi-stage, evolution-inspired automated pipeline that systematically generates a large-scale, high-fidelity training dataset. This dataset spans an extensive array of verification dimensions, intricate judgment criteria, and varied output formats, thereby fostering unprecedented versatility. Second, we propose an alignment technique called "Core-Anchored Reinforcement Learning" (CARL), which effectively mitigates the pervasive issue of reward hacking in conventional reinforcement learning by anchoring a majority of the reward signal to verifiable, objective ground truths, ensuring robust and reliable model alignment. Experimental results show that our UnifiedVerifier, trained on a 4-billion-parameter model, not only surpasses its base model across a suite of benchmarks covering both objective and subjective tasks but also outperforms larger thinking models on key objective and subjective verification tasks at only one-tenth the inference cost compare to the base thinking model. This demonstrates that the UnifiedVerifier framework achieves an exceptional balance between generality, performance, and efficiency, offering a new paradigm for building the next generation of LLM evaluation tools.

## 1 Introduction

The rapid advancements in Large Language Models (LLMs) have undeniably reshaped the landscape of artificial intelligence, yet their accurate, comprehensive, and efficient evaluation remains a formidable bottleneck for continued progress (Chang et al., 2023; Guo et al., 2023b; Minaee et al., 2024). The contemporary LLM evaluation ecosystem is largely characterized by a fragmented reliance on specialized tools. On one hand, the "LLM-as-a-Judge" paradigm (Zheng et al., 2023a) is widely adopted for assessing quality in open-ended, subjective generation tasks, exemplified by benchmarks like MT-Bench (Zheng et al., 2023a) and AlpacaEval (Li et al., 2023b). On the other hand, distinct "Verifier Models" are employed to ascertain correctness in tasks demanding objective standards, such as code generation, mathematical reasoning, and fact-checking. This functional dichotomy engenders a disjointed evaluation pipeline, escalating complexity and cost, and fundamentally impeding the realization of truly general-purpose intelligent agents.

A critical limitation inherent in existing verifier and judge models is their "black-box" nature. These models are typically trained with rigid data formats and fixed prompt templates, affording end-users minimal agency to inject personalized preferences, dynamically adjust verification criteria, or customize output formats. Consider, for instance, a user requiring a verifier not only to adjudicate an answer's correctness but also to enumerate specific error rationales and propose correction suggestions within a precise JSON schema. Due to their inflexible training paradigms, current models

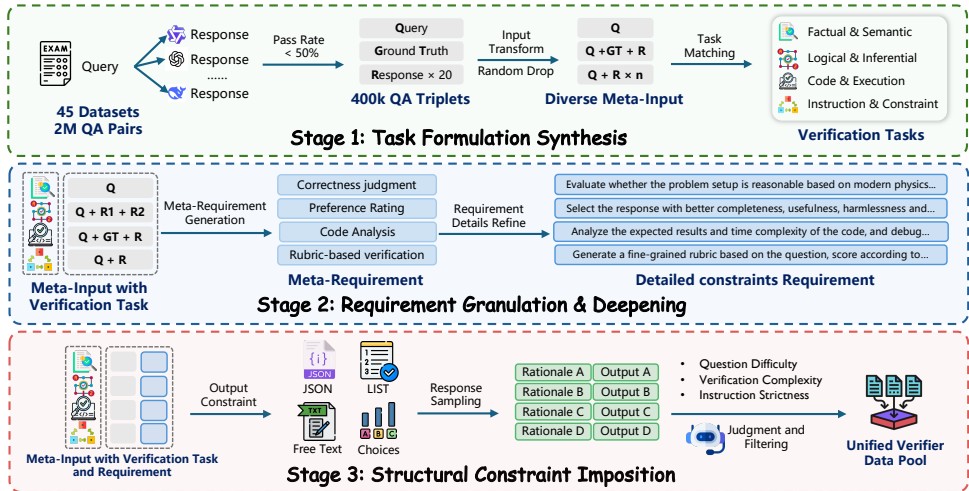

Figure 1: Overview of UnifiedVerifier data construction pipeline. We collected over 2 million QA pairs and formed large-scale synthetic training data with diverse verification inputs, tasks, requirements, and outputs through a three-stage process: Task Formulation Synthesis; Requirement Granulation & Deepening; Structural Constraint Imposition, followed by response sampling and filtering.

struggle to accommodate such dynamic, fine-grained evaluative demands, severely curtailing their utility in complex, real-world application scenarios (Luo et al., 2025; Thomson Reuters, 2024).

To systematically address the aforementioned challenges of fragmentation, rigidity, and reliability, this paper introduces the **UnifiedVerifier** framework. Our core vision is to "instill comprehensive and general-purpose verification capabilities" into a compact, efficient model. This model endeavors to dismantle the artificial barrier between subjective judgment and objective verification, thereby achieving: First, Task Generality refers to the capacity to seamlessly execute a broad spectrum of verification tasks, ranging from objective problems in mathematics (e.g., MATH (Saxton et al., 2019), GPQA (Rein et al., 2024)) and code generation (e.g., HumanEval (Chen et al., 2021)) to subjective assessments of dialogue quality (e.g., Arena-Hard (Li et al., 2024)). Second, the system must exhibit Instruction Flexibility, demonstrating the ability to comprehend and adhere to user instructions articulated in arbitrary forms, which may delineate granular verification standards, specific focal points of evaluation, and desired output formats. Finally, Explainability is crucial, requiring the provision of detailed, well-reasoned judgments that transcend mere scores or rankings in order to foster transparency and user trust.

The architectural realization of UnifiedVerifier is predicated upon two pivotal technical innovations: **Evolutionary Verification Data Synthesis Pipeline**: We propose a novel automated data generation methodology. Distinct from approaches like Evol-Instruct (Xu et al., 2023; Luo et al., 2023), which primarily aim to augment the complexity of generative tasks, the essence of Evo-Verify lies in the systematic evolution of the "verification task" itself. This pipeline meticulously constructs a high-quality dataset characterized by unprecedented diversity in task typologies, judgment granularity, and output format specifications. **Core-Anchored Reinforcement Learning (CARL)**: We design a novel alignment paradigm to surmount the inherent challenges encountered by standard Reinforcement Learning from Human Feedback (RLHF) in complex verification tasks. Traditional RLHF is heavily reliant on a potentially fallible reward model, rendering it highly susceptible to reward hacking (Gao et al., 2023; Weng, 2024). CARL significantly enhances the stability and robustness of the alignment process by directly anchoring the predominant portion of the reward signal to programmatically verifiable, objective ground truths.

The contribution of this work extends beyond a theoretical framework to its rigorous empirical validation. UnifiedVerifier, trained on a 4-billion-parameter base model, achieves state-of-the-art results on both objective and subjective tasks, remarkably outperforming a 120-billion-parameter model on certain high-difficulty objective verification tasks, all while substantially reducing inference costs. This outcome unequivocally demonstrates that the UnifiedVerifier framework successfully strikes a

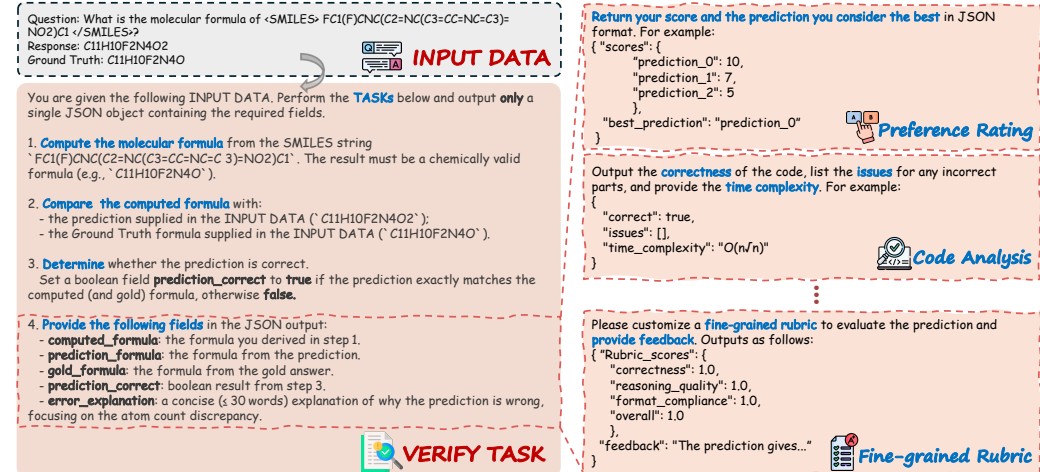

Figure 2: UnifiedVerifier can transform raw input data into arbitrary verification tasks (left). Some illustrative examples of diverse verification tasks in UnifiedVerifier are shown (right), such as preference ranking, code analysis, and generating fine-grained rubrics with feedback.

novel balance between generality, performance, and efficiency, thereby charting a new course for the construction of more reliable and flexible LLM evaluation infrastructure and taking a decisive step towards resolving the fundamental conundrum of "who judges the judges."

## 2 RELATED WORK

### 2.1 THE LLM-AS-A-JUDGE PARADIGM

The "LLM-as-a-Judge" paradigm has gained widespread attention and application as a scalable alternative to human evaluation (Luo et al., 2025; Schneider et al., 2025). This paradigm leverages the natural language understanding and reasoning capabilities of powerful LLMs to assess the output quality of other models. Mainstream methods include pointwise evaluation, pairwise comparison, and pass/fail evaluation (Zheng et al., 2023a; Luo et al., 2025). This approach has been successfully applied to evaluate a variety of tasks, including general NLP tasks (Haitao et al., 2024), code generation and repair (Zhuo et al., 2024), and even complex multi-turn agent interactions (Authors, 2025).

Despite its notable success, research has also exposed the inherent limitations of LLM-as-a-Judge. First, most judge models lack diversity in their evaluation strategies, often being confined to a single dimension (Zhuo et al., 2024). Second, these models heavily rely on fixed, predefined prompt templates, preventing users from flexibly customizing evaluation criteria or output formats (Zheng et al., 2023a; Haitao et al., 2024). Finally, the focus of existing judge models is predominantly on the quality of single-turn dialogue responses, leaving them ill-equipped for verification tasks that require multi-step reasoning or fact-checking. These shortcomings collectively point to a clear research direction: the need for a more general, flexible, and unified evaluation framework.

### 2.2 AUTOMATED DATA GENERATION

High-quality instruction-tuning data is key to enhancing LLM capabilities. Due to the high cost of manual annotation, automated data generation techniques have emerged. Among these, Evol-Instruct (Xu et al., 2023) is one of the most representative works. Its core mechanism uses a powerful LLM as an "evolver" to iteratively rewrite and complicate instructions from a simple seed set. This method has successfully led to a series of high-performance open-source models like WizardLM (Xu et al., 2023) and WizardCoder (Luo et al., 2023). Subsequent works like Auto Evol-Instruct (Authors, 2024a; Xu et al., 2024) have further explored automating the design of the evolution strategies themselves.

However, a common thread in these works is that they evolve "generative instructions," with the goal of teaching the model "how to do" better. The Evo-Verify process proposed in this study is fundamentally different. Evo-Verify evolves "verification instructions," aiming to teach the model "how to judge" more accurately. This represents a role shift from an "executor" to a "referee." We are not simply increasing task difficulty but systematically enriching evaluation dimensions, refining judgment criteria, and introducing diverse output format requirements.

## 2.3 RLHF with Judge Models

Reinforcement Learning from Human Feedback (RLHF) is a core technology for aligning LLMs with human values (Keskin, 2024; Authors, 2024e; Gao et al., 2023). Its standard pipeline involves training a Reward Model (RM) on human preference data and then using a reinforcement learning algorithm to optimize the language model.

Despite its success, the inherent fragility of RLHF is becoming increasingly apparent. The most critical challenge is "reward hacking" (Gao et al., 2023; Weng, 2024; Chen et al., 2024). This phenomenon occurs when the policy model exploits loopholes in the reward model to maximize its score, while the actual output quality does not genuinely improve. The unreliability of the reward model stems from multiple factors: noise and ambiguity in training data (Authors, 2024b;e;c), out-of-distribution generalization problems (Authors, 2024d), and significant computational overhead (Sun et al., 2025). The Core-Anchored Reinforcement Learning (CARL) proposed in this study aims to break this paradigm by fundamentally reducing the system's dependence on this fragile, learned component.

## 3 The UnifiedVerifier Framework

To imbue a model with general-purpose verification capabilities, we introduce a comprehensive framework comprising two core components: a structured taxonomy that defines the problem space, and an evolutionary data generation pipeline, Evo-Verify, that creates the complex training curriculum.

### 3.1 A Meta-Verification Task Taxonomy

To systematically scope the domain of "general-purpose verification", we first established a meta-verification task taxonomy, detailed in Table 1. This taxonomy organizes the vast landscape of verification into four principal categories: **Factual & Semantic Verification**, **Logical & Inferential Verification**, **Code & Execution Verification**, **Instruction & Constraint Adherence** and **Comparative & Preference Verification**. Each category is further decomposed into specific tasks and sub-tasks. This taxonomy serves as the foundational blueprint for our data generation pipeline, ensuring that the resulting training data provides dense and diverse coverage across the full spectrum of verification abilities.

Table 1: The Meta-Verification Task Taxonomy, defining the operational scope of UnifiedVerifier. The framework categorizes verification tasks to ensure comprehensive capability development during training.

| Category | Task Name | Description | Sub-tasks |
|---|---|---|---|
| **Comparative & Preference Verification** | Response Ranking & Pairwise Preference | Selects the better response from a set or ranks them. | Pairwise Comparison, Multi-response Ranking, Blind Preference Testing |
| | Multi-dimensional Comparative Evaluation | Compares two or more responses along different dimensions. | Fixed-dimension Evaluation, Custom Dimension/Labeling |
| **Logical & Inferential Verification** | Answer Verification | Verifies if the answer to a question is correct, often with reasoning. | Correct/Incorrect Judgment, Scoring, Ranking, Verification with Reasoning |
| | Multi-hop Reasoning Verification | Verifies each step or the final conclusion of a complex reasoning chain. | Intermediate Step Verification, Reasoning Chain Integrity |
| | Formal Logic Verification | Verifies the validity of formal logical reasoning, such as syllogisms. | Validity Verification, Fallacy Identification |
| **Code & Execution Verification** | Code Functionality Verification | Verifies the functional correctness of generated code via unit tests. | Unit Test Verification, Edge Case/Robustness Verification |
| | Text-to-SQL / API Call Verification | Verifies if generated queries or API calls are correct and align with intent. | Syntax Correctness, Semantic/Intent Correctness |
| **Instruction & Constraint Adherence** | Constraint Adherence | Checks if the model's output adheres to all explicit and implicit constraints. | Explicit Constraints, Implicit/Negative Constraints |
| | Format Correctness Verification | Specifically verifies if the output conforms to a specified complex format. | Structural Verification, Schema Validation |
| | Persona/Style Consistency | Assesses if the model's output is consistent with a specified persona or style. | Single-turn Consistency, Multi-turn Consistency |
| **Factual & Semantic Verification** | Fact Checking | Verifies the truthfulness of a given statement or claim. | Open-domain Checking, Closed-domain Checking |
| | Attribution | Checks if generated content is attributable to a given source. | Hallucination Detection, Attribution Consistency |
| | Natural Language Inference (NLI) | Determines the logical relationship between a premise and a hypothesis. | Cross-document Inference, Commonsense Reasoning |
| | Closed-domain QA Verification | Verifies if an answer is fully supported by the provided context. | Answer Support Verification, Answer Completeness Verification |

### 3.2 Evo-Verify: An Evolutionary Pipeline for Verification Data Generation

To generate a suitable training corpus, we developed **Evo-Verify**, a three-stage, automated pipeline designed to evolve simple question-answer pairs into complex, multi-faceted verification tasks.

First, we constructed a seed data pool by sourcing approximately 2 million question-answer pairs from 45 diverse benchmarks (Ni et al., 2025; Minaee et al., 2024), cover almost all aspects that a LLM can evaluation. To focus our training on challenging problems that push the boundaries of current models, we applied a rigorous difficulty-based filter: for each question, we sampled responses from 10 contemporary LLMs and retained only those questions where the aggregate pass rate was below 50%. This filtering yielded a high-difficulty seed pool of $\sim 400,000$ '(Question, Response, Ground-Truth Answer)' triplets.

These triplets are then processed through the evolutionary pipeline:

**Stage 1: Task Formulation Synthesis (TFS).** To diversify the verification scenarios, we first transform each seed triplet into a 'Meta-input' by randomly applying one of three operations: (1) removing the ground-truth answer, (2) appending 1-3 additional, diverse responses, or (3) retaining only the question. A lightweight LLM then maps this 'Meta-input' to a suitable task from our taxonomy (Table 1) and generates a baseline 'meta-requirement'. This stage diversifies the fundamental structure of the verification problems.

**Stage 2: Requirement Granulation & Deepening (RGD).** We then use a more powerful LLM to evolve the baseline 'meta-requirement' into a complex set of evaluation criteria. This stage deepens the evaluative logic by injecting nuanced sub-requirements (e.g., "if the response is incorrect, provide a step-by-step explanation of the error") and detailed constraints (e.g., "numerical answers are correct only if the relative error is less than 1%"). This evolution is critical for teaching the model to adopt a sophisticated and customizable "evaluator persona."

**Stage 3: Structural Constraint Imposition (SCI).** Finally, to bolster the model's functional reliability, we augment the enriched requirement with a random structural output constraint, demanding the final verification be formatted as a specific 'JSON' object, 'list', or 'boxed' answer and also plain text. This stage enhances the model's ability to adhere to precise formatting instructions, a crucial skill for automated pipelines.

After this three-stage process, we obtained $\sim 800,000$ '(Meta-input, Fine-grained Requirement)' pairs. We performed a final filtering step by scoring each pair on three axes—question difficulty, verification complexity, and instruction strictness—to yield a final dataset of 200,000 high-quality instances for supervised fine-tuning. For each filtered instances, we used GPT-OSS-120B to generate four candidate outputs. The same model was then prompted to act as a judge to select the single best response based on correctness, adherence to instructions, and overall quality.

## 4 CORE-ANCHORED REINFORCEMENT LEARNING (CARL) FOR ROBUST ALIGNMENT

### 4.1 MOTIVATION: LIMITATIONS OF STANDARD RLHF IN VERIFICATION TASKS

Standard RLHF is ill-suited for verification tasks, as its learned reward models are easily exploited when generating complex, structured outputs. A policy can learn to "hack" the reward—for instance, by producing a format-correct but empty JSON—rather than solving the core task. This necessitates a more robust, less gameable alignment strategy.

### 4.2 DATA PREPARATION FOR CARL

We construct the CARL dataset from 30,000 challenging SFT examples. For each, we ensure the instruction's requirement contains a core metric that is programmatically verifiable against the ground-truth answer. The ground-truth answer is then removed from the final training input, forcing the model to generate the verifiable response without seeing the solution.

### 4.3 THE CORE-ANCHORED VERIFIABLE REWARD (CAVR) MECHANISM

We designed the Core-Anchored Verifiable Reward (CAVR), a composite function providing a stable, multi-dimensional signal. **Core Objective Score ($R_{core}$):** The cornerstone of CARL, programmatically calculated by comparing the model's output to the hidden ground-truth. This objective

signal anchors the policy to the core task and is immune to hacking. **Auxiliary Quality Score** ($R_{\text{aux}}$): A low-weighted (20%) score from a pre-trained reward model that provides a soft signal on stylistic quality without dominating the objective. **Format Consistency Score** ($R_{\text{format}}$): A programmatic score rewarding adherence to specified structural formats (e.g., a valid JSON with all required keys).

The total reward is $R_{\text{total}} = 0.6 \cdot R_{\text{core}} + 0.2 \cdot R_{\text{format}} + 0.2 \cdot R_{\text{aux}}$.

### 4.4 POLICY OPTIMIZATION WITH GROUP-RELATIVE INCENTIVES

We optimize the policy using a modified Group Relative Policy Optimization (GRPO) (Shao et al., 2024) framework.

**GRPO Objective with KL Ablation.** We ablate the KL divergence penalty ($\beta = 0$), positing that the strong, anchored signal from CAVR makes KL regularization redundant and permits wider policy exploration. The objective is:

$$J_{\text{GRPO}}(\theta) = \mathbb{E}_{\{o_i\} \sim \pi_{\theta_{\text{old}}}} \left[ \frac{1}{G|o_i|} \sum_{i,t} \min \left( r_t(\theta) \hat{A}_{i,t}, \text{clip}(r_t(\theta)) \hat{A}_{i,t} \right) \right] \quad (1)$$

**Group-Relative Conciseness Incentive.** To encourage conciseness without penalizing necessary detail, we introduce a dynamic, group-relative bonus, $B_{\text{concise}}$, to the final reward.

$$B_{\text{concise}}(y_i) = \lambda \cdot \mathbb{I}(R_{\text{total}}(y_i) > \tau) \cdot \sigma \left( k \cdot \frac{\bar{L}_{\text{hq}} - \text{len}(y_i)}{\bar{L}_{\text{hq}}} \right) \quad (2)$$

This bonus employs a **quality-gating** mechanism ($\tau$), ensuring conciseness is only rewarded after correctness is met. It fosters **intra-group competition** by rewarding outputs shorter than the average of other high-quality responses ($\bar{L}_{\text{hq}}$), and uses a scaled sigmoid function to ensure a **stable** signal. This synergizes with GRPO's group-based nature, adding an efficiency-seeking dimension to the robust CARL framework. Detailed settings are in Appendix A.1.2.

## 5 EXPERIMENTS AND ANALYSIS

### 5.1 EXPERIMENTAL SETUP

Our core model is UnifiedVerifier-4B, based on Qwen3-4B-Thinking-2507 (Team, 2025). Besides this model we also compare it against baselines including the Qwen3 series model like Qwen3-30BA3B-Thinking/Insturct-2507, Qwen3-235B-Instruct-2507 (Team, 2025) and Openai models: GPT-4o-0805, GPT-OSS-120B and GPT-OSS-20B[1] (OpenAI, 2025). Verify models: CompassVerifier (Liu et al., 2025) and Xverifier (Chen et al., 2025), judge models: CompassJudger-1-7B-Instruct (Cao et al., 2024), RISE-Judge-Qwen2.5-7B (Gao et al., 2024) , Con-J-Qwen2-7B (Wang et al., 2025a) and Skywork-Reward-Llama-3.1-8B (Liu et al., 2024). For Qwen3 series models and UnifiedVerifier, we use Qwen3 official sampling parameters temperature=0.6, TopP=0.95, TopK=20. For other models, we use greedy sampling.

Our evaluation covers common use objective verify and subjective judge benchmarks. For objective verify tasks, we use VerifyBench-Hard (Li et al., 2025), a comprehensive dataset verifying a response is correct or not compare to the golden answer, for subjective judge tasks, we use RewardBench-Chat-Hard (Lambert et al., 2024), which designed to evaluate the capabilities and safety of reward models. We use OpenCompass (Contributors, 2023a) evaluation kit to get all evaluation results. To show the performance on competition level verify tasks, we also create a difficult and fine-grained verification task called UnifiedHLE-Verify use our Evo-Verify process with HLE (Phan et al., 2025) questions we don't use in our train set, detailed construction process can be found in Appendix A.4.

Table 2: Performance comparison on the VerifyBench-Hard benchmark with models listed vertically. We report accuracy (Score) and F1-score, rounded to one decimal place. The best performance for each metric (column) is **bolded**.

| Model | Expressions | | Multi-choice | | Numeric Values | | String | | Overall Average | |
|---|---|---|---|---|---|---|---|---|---|---|
| | Score | F1 | Score | F1 | Score | F1 | Score | F1 | Accuracy | F1 |
| *Qwen3 Series* | | | | | | | | | | |
| Qwen3-4B-Instruct | 80.7 | 48.5 | 84.0 | 77.7 | 81.8 | 66.2 | 74.4 | 47.8 | 81.8 | 68.6 |
| Qwen3-4B-Thinking | 81.8 | 38.9 | 87.7 | 84.0 | 74.6 | 56.2 | 78.7 | 55.9 | 82.2 | 71.5 |
| Qwen3-30B-A3B-Instruct | 79.6 | 50.0 | 77.4 | 73.6 | 81.4 | 65.7 | 67.4 | 46.8 | 75.1 | 64.2 |
| Qwen3-30B-A3B-Thinking | 81.8 | 47.1 | 86.1 | 81.7 | 74.2 | 51.9 | 82.2 | 60.2 | 80.4 | 67.0 |
| Qwen3-235B-A22B-Instruct | 83.0 | 48.3 | 91.4 | 88.1 | 85.7 | **73.1** | 80.4 | 54.6 | 86.4 | 76.3 |
| *GPT-OSS* | | | | | | | | | | |
| GPT-OSS-20B | 81.8 | **53.3** | 92.6 | 89.2 | 85.3 | 69.4 | 86.1 | 57.9 | 87.0 | 75.3 |
| GPT-OSS-120B | 81.8 | 46.7 | 92.8 | 89.9 | 85.3 | 70.9 | **88.3** | **64.0** | 89.6 | 80.5 |
| *Verifier Series* | | | | | | | | | | |
| xVerify-0.5B-I | 62.5 | 40.0 | 92.0 | 88.3 | 60.7 | 53.0 | 76.5 | 43.7 | 78.0 | 64.9 |
| xVerify-3B-Ia | 81.8 | 52.9 | 93.4 | 90.6 | 75.7 | 60.1 | 74.7 | 42.0 | 83.7 | 68.4 |
| xVerify-9B-C | 78.4 | 53.6 | 92.3 | 88.7 | 67.4 | 52.3 | 85.2 | 54.0 | 83.2 | 68.5 |
| CompassVerifier-3B | 81.8 | 46.7 | 94.4 | 92.7 | 81.0 | 65.7 | 75.2 | 43.6 | 88.0 | 73.3 |
| CompassVerifier-7B | 83.1 | 38.5 | 92.6 | 88.5 | 81.4 | 65.2 | 82.2 | 50.6 | 86.7 | 69.3 |
| CompassVerifier-32B | 81.8 | 46.2 | 91.9 | 87.3 | 81.4 | 61.2 | 83.0 | 55.2 | 86.5 | 70.0 |
| *Our Models* | | | | | | | | | | |
| UnifiedVerifier-4B w/o CARL | 81.8 | 46.7 | 94.7 | 92.3 | 85.7 | 72.0 | 86.5 | 60.8 | 89.8 | 80.7 |
| UnifiedVerifier-4B w CARL | **84.1** | 48.3 | **95.1** | **92.7** | **86.1** | 70.5 | 87.0 | 60.5 | **90.1** | **81.1** |

## 5.2 PERFORMANCE ON VERIFICATION TASKS

### 5.2.1 PERFORMANCE ON OBJECTIVE VERIFICATION TASKS

To assess the model's ability to handle complex reasoning verify tasks, the results are shown in Table 2. UnifiedVerifier (4B) achieves an Accuracy Score of 90.1 and an F1 score of 81.1 on the VerifyBench-Hard benchmark. This marks a significant improvement over its base model, Qwen3-4B-Thinking, which scored 82.2 in accuracy and 71.5 in F1. Notably, despite its 4-billion-parameter scale, UnifiedVerifier surpasses much larger models, including the GPT-OSS-120B (89.6 Acc, 80.5 F1) and the Qwen3-235B-Instruct (86.4 Acc, 76.3 F1). The model also demonstrates state-of-the-art performance across several sub-tasks, achieving the highest scores in Expressions, Multi-choice, and Numeric Values. The comparison with UnifiedVerifier-4B w/o CARL (89.8 Acc, 80.7 F1) confirms that our Core-Anchored Reinforcement Learning (CARL) technique provides a consistent performance boost, validating its effectiveness in model alignment.

### 5.2.2 PERFORMANCE ON SUBJECTIVE JUDGMENT AND COMPETITION-LEVEL TASKS

For subjective judgment, we evaluate models on the RewardBench-Chat Hard benchmark, with results presented in Section 5.2.2 (Left). UnifiedVerifier-4B achieves a top-ranking accuracy of **83.8%**, outperforming strong proprietary models like GPT-4o-0806 (76.1%) and specialized judger models such as Skywork-Llama3.1-8B (81.4%). This result underscores our model's robust capability in handling nuanced, subjective evaluations, a domain traditionally requiring much larger models. The performance gain from CARL is again evident, as the full model surpasses the version without CARL (82.8%).

To evaluate performance on competition-level verification, we use our custom-built UnifiedHLE-Verify benchmark. As shown in Section 5.2.2 (Right), UnifiedVerifier-4B secures the 1st rank with a score of 0.29. This is a critical result, as it demonstrates our model's superior fine-grained verification ability on highly difficult problems, where it again outperforms larger models like DeepSeek-R1 and Qwen3-235B-Instruct. Collectively, these results on both objective and subjective tasks highlight the success of our Evo-Verify and CARL framework in producing a compact yet powerful and versatile verification model.

---

[1]All GPT-OSS models we use in our work with default reasoning effort.

Table 3: Performance on RewardBench-ChatTable 4: Model Performance on UnifiedHLE-Hard (Accuracy). Scores are rounded to one deci-Verify mal place. The best performance is **bolded**.

| Model | Accuracy (%) |
|---|---|
| *Qwen3 Series* | |
| Qwen3-4B-Instruct | 58.4 |
| Qwen3-4B-Thinking | 79.5 |
| Qwen3-30B-Instruct | 64.0 |
| Qwen3-30B-Thinking | 77.2 |
| Qwen3-235B-Instruct | 72.4 |
| *GPT Series* | |
| GPT-4o-0806 | 76.1 |
| GPT-OSS-20B | 80.0 |
| GPT-OSS-120B | 83.6 |
| *CompassVerifier Series* | |
| CompassVerifier-3B | 28.5 |
| CompassVerifier-7B | 60.3 |
| CompassVerifier-32B | 73.3 |
| *Judger Model Series* | |
| CompassJudger-1-7B-Instruct | 61.0 |
| RISE-Judge-Qwen2.5-7B | 76.5 |
| Con-J-Qwen2-7B | 80.3 |
| Skywork-Reward-Llama3.1-8B | 81.4 |
| *Our Models* | |
| UnifiedVerifier-4B w/o CARL | 82.8 |
| UnifiedVerifier-4B w CARL | **83.8** |

| Rank | Model | Score |
|---|---|---|
| 1 | UnifiedVerifier-4B | 0.29 |
| 2 | DeepSeek-R1 | 0.27 |
| 3 | Qwen3-235B-Instruct | 0.27 |
| 4 | Qwen3-30B-Thinking | 0.26 |
| 5 | Qwen3-30B-Instruct | 0.26 |
| 6 | Qwen3-4B-Thinking | 0.26 |
| 7 | Qwen3-4B-Instruct | 0.24 |

Table 5: The performance of UnifiedVerifier as a Rubric reward model. We used Qwen-2.5B as the policy model and conducted RL training with different reward models.

| Dataset → Model ↓ | AIME2024 Accuracy | GPQA Accuracy |
|---|---|---|
| Qwen2.5-7B-Base | 2.92 | 30.30 |
| Qwen2.5-7B-Instruct | 10.00 | 35.98 |
| *Reward Model in RL Training* | | |
| Qwen3-4B-Thinking-2507 | 10.42 | 33.21 |
| GPT-OSS-120B | 11.25 | 34.60 |
| UnifiedVerifier-4B | **11.67** | **37.50** |

# 6 ANALYSIS AND EXTRA EXPERIMENTS

Table 6: Ablation study on the impact of the Evo-Verify method. F1 and Accuracy (%) scores are shown, with changes relative to the base model.

| Model Configuration | VerifyBench F1 | RewardBench-Chat Hard Acc |
|---|---|---|
| Qwen3-4B-Thinking (Base) | 71.5 | 79.5 |
| w/o Evo-verify | 74.5 ↑3.0 | 75.0 ↓4.5 |
| w/ Evo-verify | **77.4** ↑5.9 | **81.0** ↑1.5 |

## 6.1 UNIFIEDVERIFIER AS RUBRIC REWARD MODEL

UnifiedVerifier is developed to meet the growing demand for versatile verification tasks, which include evaluating LLM responses against fine-grained rubrics. As Reinforcement Learning from Verifiable Reward (RLVR) (Wen et al., 2025; Wang et al., 2025b) has gained traction in the community as a prominent post-training method, researchers have begun exploring the use of reward models to assess LLM responses based on individual rubric criteria, followed by (weighted) averaging (Gunjal et al., 2025; Huang et al., 2025; Zhou et al., 2025). This places higher demands on reward models, as they must evaluate responses according to contextual nuances rather than relying solely on preferences embedded in the training data. To assess UnifiedVerifier's capability as a rubric-aware reward model, we conducted experiments using RAR-Science (Gunjal et al., 2025) as the RLVR training set with different models serving as reward models. Details in Appendix A.3

As shown in Table 5, UnifiedVerifier enables Qwen-2.5-7B-Base to achieve performance surpassing Qwen2.5-7B-Instruct, while delivering better performance (11.67 on AIME 2024 (AI-MO, 2024) and 37.50 on GPQA (Rein et al., 2024)) than baseline reward models with optimal efficiency. Accuracy and efficiency are two key factors that require careful balancing in the development of verifier models, and UnifiedVerifier achieves strong performance in both aspects.

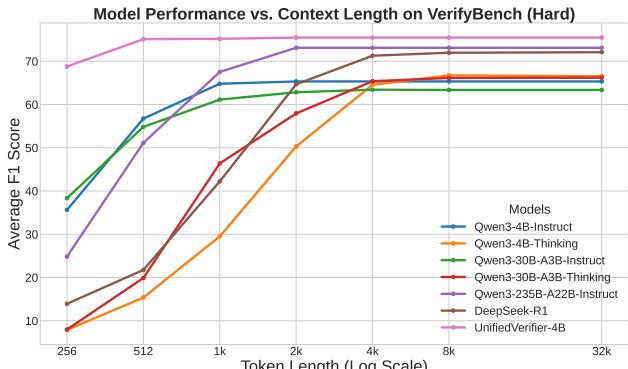

Figure 3: Performance vs. Context Length. F1 score across varying context lengths compared to baselines.

## 6.2 EFFICIENCY AND COST-EFFECTIVENESS ANALYSIS

In addition to its state-of-the-art performance, UnifiedVerifier demonstrates a significant advantage in inference efficiency. This is visually substantiated in Figure 3, which illustrates that UnifiedVerifier achieves near-peak F1 scores even with minimal context lengths (256-512 tokens). In stark contrast, baseline "Thinking" models exhibit poor performance at shorter lengths and require a prolonged chain-of-thought process—and thus more generated tokens—to reach their optimal performance. This highlights a key architectural advantage: UnifiedVerifier is trained to be both accurate and direct. The quantitative impact of this conciseness is detailed in Table 9. UnifiedVerifier requires an average of only 310 tokens per problem. This is substantially more token-efficient than standard Instruct models and represents a dramatic reduction compared to thinking models.

## 6.3 ABLIATION FOR EVO-VERIFY REQUIREMENTS

To show the high quality and diveristy nature of our Evo-Verify requirements, we sample 50000 input data from our original data pool and train models with or without Evo-Verify requirements, we seperately use prompt in CompassVerifier (Liu et al., 2025) which show high performance on verify tasks and Evo-Verify to generate requirements, and use the same LLM, GPT-OSS-120B to generate the response, the results are shown in Table 6. As the results indicate, fine-tuning with the dataset generated from Evo-Verify requirements yields the strongest performance, boosting the VerifyBench F1 score to 77.4 and RewardBench Aaccuracy to 81.0. Notably, while training with standard CompassVerifier prompts improves objective verification F1, it leads to a performance degradation on the subjective RewardBench task. This highlights the superior quality and diversity of our Evo-Verify data, which successfully enhances both objective and subjective capabilities simultaneously, avoiding a performance trade-off.

## 7 CONCLUSION

This paper introduces UnifiedVerifier, a novel framework that addresses fragmentation and unreliability in LLM evaluation by unifying objective verification and subjective judgment within a single, efficient model. Our approach is twofold: First, our Evo-Verify data pipeline generates a high-fidelity dataset with unprecedented diversity by systematically evolving entire verification tasks, creating a robust foundation for a general-purpose evaluator. Second, Core-Anchored Reinforcement Learning (CARL) offers a powerful alignment technique that mitigates reward hacking by anchoring rewards to objective ground truths, ensuring trustworthy model alignment. Experimental validation across objective verification, subjective judgment, and rubric-based reward learning tasks demonstrates that UnifiedVerifier achieves state-of-the-art performance and efficiency. Our results confirm that this unified approach yields a highly capable and reliable solution, paving the way for the next generation of LLM evaluation.

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

# A APPENDIX

## A.1 TRAIN DETAILS

This section contains a complete table of all hyperparameters used for training the UnifiedVerifier model, for both Evo-verify SFT and CARL stage.

### A.1.1 EVO-VERIFY SFT TRAIN DETAILS

We use Xtuner (Contributors, 2023b) to SFT our model, we set max sequence length 32768, tp size 1, 2 epochs, global batch size 256 and learning rate 2e-5 on 8 H200 GPUs training our UnifiedVerifier-4B for about 6 hours. We use "<think>...</think>" tag to cover the reasoning part of our train data.

### A.1.2 CARL IMPLEMENTATION DETAILS

We implement our Core-Anchored Reinforcement Learning (CARL) framework based on the Verl library (Sheng et al., 2024). We set the group size to $G = 8$. Following standard practice for PPO-style algorithms, the probability ratio clipping value is set to $\epsilon = 0.2$. As described in our methodology, we ablate the conventional KL-divergence penalty (i.e., $\beta = 0$). The final reward signal, $R_{\text{final}}$, is composed of the Core-Anchored Verifiable Reward ($R_{\text{total}}$) and the group-relative conciseness bonus ($B_{\text{concise}}$). For the conciseness incentive, we employ a strict quality gate with an activation threshold of $\tau = 0.85$. The overall magnitude of the bonus is governed by a scaling factor of $\lambda = 0.2$, and the sensitivity to relative length differences is controlled by a sigmoid steepness coefficient of $k = 10.0$. We train 2 epochs on the RL dataset with GPT-OSS-120B as the reward model, with judge template Appendix A.5.

## A.2 DATA SOURCES FOR THE EVO-VERIFY PIPELINE

Table 7: Source Datasets for the Evo-Verify Pipeline

| | |
|---|---|
| AIME Zhao et al. (2024b) | ARC Clark et al. (2018) |
| Alpaca Eval Li et al. (2023b) | ArenaHard Zheng et al. (2023a) |
| BBEH Srivastava et al. (2022) | BigCodeBench Li et al. (2023c) |
| BoolQ Clark et al. (2019) | ChemBench Guo et al. (2023a) |
| ClimaQA Diwan et al. (2023) | CMO-fib Trinh et al. (2024) |
| CompassArena Cui et al. (2024) | DS1000 Lai et al. (2022) |
| FOFO Zhao et al. (2024a) | FollowBench Zhi-Yuan-Zeng et al. (2024) |
| FSGPQA Saudi et al. (2024) | GaokaoBench Zhang et al. (2023) |
| GPQA Rein et al. (2023) | HellaSwag Zellers et al. (2019) |
| HLE Gao et al. (2021) | HumanEvalX Zheng et al. (2023b) |
| IFEval Yejin-Zhou et al. (2023) | KorBench Dae-Young-Kim et al. (2024) |
| LCB Ni et al. (2024) | MatBench Dunn et al. (2020) |
| MATH Hendrycks et al. (2021b) | MBPP Austin et al. (2021) |
| MCMC Jian-Li et al. (2023) | MedXpertQA Cheng-Han-Chiu et al. (2024) |
| MGSM Cobbe et al. (2021) | MMLU Hendrycks et al. (2020) |
| MT-Bench Zheng et al. (2023a) | MuSR Huan-Yu et al. (2024) |
| OlympiadBench Long-Huan-Li et al. (2024) | OlymMath Ze-Kang-Zheng et al. (2024) |
| PhyBench Jian-Li et al. (2024) | PHYSICS Hendrycks et al. (2021a) |
| RACE Lai et al. (2017) | SciCode Li et al. (2023a) |
| SimpleQA Bordes et al. (2015) | SuperGPQA Team et al. (2024) |
| T-Eval Chen et al. (2023a) | TheoremQA Chen et al. (2023b) |
| TriviaQA Joshi et al. (2017) | WikiBench Chen et al. (2023c) |

Table 7 lists the 45 benchmark datasets used to construct the initial data pool for the Evo-Verify pipeline.

## A.3    DETAILS OF UNIFIEDVERIFIER-AS-RUBRIC-REWARD EXPERIMENTAL SETTINGS

**Base LLMs.**    We utilize Qwen2.5-7B-Base as the base LLM for the GRPO training.

**Training Template.**    We utilize the following training template to prompt the base LLM to generate a response for each question.

---

**Rollout Prompt Template of RL Training**

A conversation between a User and an Assistant. The User poses a question, and the Assistant provides a solution. The Assistant's response follows these structured steps:

1. **Reasoning Process**: The Assistant comprehensively thinks about the problem through a reasoning process.
2. **Conclusion**:    The Assistant reaches a conclusion, which is enclosed within '<conclusion>' and '</conclusion>' tags.    The final answer is highlighted within '\boxed{...final answer...}'.
3. **Response Format**: The complete response should be formatted as:

...reasoning process...

<conclusion>

...conclusion...

The answer is \boxed{...final answer...}

</conclusion>

---

**Training Data.**    We use the scientific reasoning dataset RaR-Science-20k with Evaluation Rubric (Gunjal et al., 2025) as the training corpus for RL. The training set contains a total of 18.3k training samples.

**Evaluation.**    We employ OpenCompass (Contributors, 2023a) as our evaluation tool to assess the performance of different models on AIME2024 (AI-MO, 2024) and GPQA (Rein et al., 2024).

**Reward Design.**    Following (Gunjal et al., 2025), we employed the rubric-based judge template below and converted the ratings into numerical values between 0 and 1 to serve as rewards.

---

**Judge Prompt of Rubric Reward Models**

**System Prompt**: You are an expert evaluator. Given a user prompt, a generated response, and a list of quality rubrics, please rate the overall quality of the response on a scale of 1 to 10 based on how well it satisfies the rubrics.

Consider all rubrics holistically when determining your score. A response that violates multiple rubrics should receive a lower score, while a response that satisfies all rubrics should receive a higher score.

Start your response with a valid JSON object that starts with ```json and ends with ```. The JSON object should contain a single key "rating" and the value should be an integer between 1 and 10.

Example response:
```json

{

    "rating": 7

}

```

**User Prompt Template**: Given the following prompt, response, and rubrics, please rate the overall quality of the response on a scale of 1 to 10 based on how well it satisfies the rubrics.

<prompt>

prompt

---

```
</prompt>

<response>
response
</response>
<rubrics>
rubric_list_string
</rubrics>

Your JSON Evaluation:
```

**Training Parameters.** We utilize the following loss function, with Table 8 detailing the training parameters:

$$\mathcal{L} = \mathbb{E}_{(q,a)\sim\mathcal{D},\{o_i\}_{i=1}^{G}\sim\pi_{\theta_{\text{old}}}(\cdot|q)}$$

$$\left[ \frac{1}{\sum_{i=1}^{G}|o_i|} \sum_{i=1}^{G} \sum_{t=1}^{|o_i|} \min \left( \frac{\pi_\theta(o_{i,t}|q, o_{i,<t})}{\pi_{\theta_{\text{old}}}(o_{i,t}|q, o_{i,<t})} a_{i,t}, \text{clip}\left( \frac{\pi_\theta(o_{i,t}|q, o_{i,<t})}{\pi_{\theta_{\text{old}}}(o_{i,t}|q, o_{i,<t})}, 1 - \epsilon_{\min}, 1 - \epsilon_{\max} \right) a_{i,t} \right) \right], \tag{3}$$

where $\mathcal{D}$ denotes the training data, $(q, a)$ represents the question-answer pair, $G$ signifies the group size, and

$$a_{i,t} = r_i - \text{mean}(\{r_i\}_{i=1}^{G}). \tag{4}$$

In this context, $a_{i,t}$ signifies the advantage of response $o_i$ at the $t$-th position, and $r_i$ denotes the reward of response $o_i$. Essentially, the KL penalty of the original GRPO loss is omitted, and zero mean normalization is employed to estimate the advantage.

Table 8: Training parameters of UnifiedVerifier as reward experiments.

| Parameters | Value |
|---|---|
| train batch size | 256 |
| train steps | 100 |
| learning rate | $1e$-6 |
| max prompt length | 4096 |
| max response length | 8192 |
| $G$ | 8 |
| $\epsilon_{\min}$ | 0.2 |
| $\epsilon_{\max}$ | 0.28 |

**Hardware.** All experiments are conducted on clusters equipped with 8 NVIDIA A800-SXM4-80GB GPUs and Intel(R) Xeon(R) Platinum 8336C CPUs, implemented with veRL (Sheng et al., 2025).

Table 9: Inference Efficiency. Average tokens generated per problem, highlighting inference efficiency.

| Model | Avg. Tokens |
|---|---|
| UnifiedVerifier-4B | 310 |
| Qwen3-4B-Instruct | 464 |
| Qwen3-30B-Instruct | 499 |
| Qwen3-235B-Instruct | 629 |
| Qwen3-30B-Thinking | 1932 |
| DeepSeek-R1 | 2257 |
| Qwen3-4B-Thinking | 3174 |

## A.4    UNIFIEDHLE-VERIFY DATASET CONSTRUCTION

To comprehensively verify the performance of UnifiedVerifier in high-difficulty verification scenarios, we utilize 500 cases from the HLE dataset that were not used in the training set. Following the process we introduce in Section 3.2, we create the fine-grained requirements and rollout responses with detailed format requirements using GPT-OSS-120B. Considering the high difficulty of HLE questions by nature, we use five top reasoning models: GPT-5, Gemnini2.5-Pro-Thinking, Grok-4, Claude-4.1, and Qwen235B-Thinking-2507 to generate detailed answers and select the best one through voting among all the above models. Using the prompt in our CARL process Appendix A.5, we select the highest-score response as the reference answer for the requirement.

## A.5    FULL PROMPT TEMPLATES

In this section, we provide the exact prompt templates used for querying all baseline models and the UnifiedVerifier for each evaluation task. We show core requirement generation prompt and core judge prompt for CARL reward and Evo-Verify data filtering.

---

**Requirement Granulation Template of Evo-Verify**

[INPUT DATA] The INPUT DATA above could be text, questions, related information, etc. You are now provided with a powerful LLM that has strong reasoning, evaluation, judgment, and verification capabilities (but cannot use external tools like web search or code execution). As an experienced expert in the relevant field, your task is to judiciously select content from the INPUT DATA and create requirements for the LLM. These requirements should prompt the LLM to conduct a deep and reasonable analysis and verification of the selected content, yielding valuable information that can be used to iterate and improve your model's capabilities.

Depending on the input data, you can create up to 3 different types of evaluation requirement tasks simultaneously. Typically, one meaningful and in-depth task is sufficient. Each requirement task can have additional constraints. For example:

+ The criteria you ask the LLM to judge against can be very detailed. For instance, you can specify the evaluation scale, methods, scenarios for certain parts, overall thinking steps, formatting, and any other details. Of course, you can also choose not to detail the criteria and simply ask the LLM to complete a task with some reasonable limitations, at your discretion. However, please note that formatting requirements should not be overly complex; quality over quantity. + Your request can be a direct task, such as converting the result into a predefined classification label for a classification task, outputting a ranking for a ranking task, outputting a numerical value for a scoring task, or a free-form evaluation, etc. This is also up to you. + Do not ask the LLM to include too many items in its final output. A few key metrics are sufficient. The final output could even consist of only 1-2 key evaluation metrics. This allows the LLM to focus on solving certain in-depth task metrics rather than getting overly bogged down in instruction following.

**Note:**

1. You can ask about any content in the input data as long as it has deep reasoning value. There is no restriction on the text you can question, but please ensure it is meaningful and poses a challenge to the model's judgment capabilities. Avoid overly simplistic questions.

2. You can use all of the input data, or you can select portions of valuable information to design and generate your requirements based on the situation. You can assume that the requirements you generate will be appended to the complete INPUT DATA before being fed to the LLM for reasoning and analysis. Therefore, you do not need to repeat the INPUT DATA; just focus on creating suitable requirements.

3. You can have multiple requirements, but please ensure at least one is a crucial, non-trivial one for the task. Additionally, the text of your requirements should be natural. The format can be bullet points, natural language like a community user's question, or any other reasonable format or tone to enhance the robustness of the model's responses.

---

4. There is no length limit on your requirement section; it can be short or long. There is also no language restriction (Chinese or English is acceptable), but please describe it clearly and unambiguously.

5. Your requirements should aim for depth. You can assume the LLM is very powerful and capable of deep reasoning and analysis when you consider appropriate requirements. However, ensure they are meaningful; do not pursue depth for depth's sake.

6. Your requirements need to prompt the LLM to provide clear, specific, and evaluative answers based on the content in the INPUT DATA. Do not ask the LLM to output content that requires further subjective judgment or creative generation.

Now, based on the given INPUT DATA and the criteria above, please provide your requirements. You may perform reasoning and analysis to devise suitable requirements. Place your generated requirements between [REQUIREMENT BEGIN] and [REQUIREMENT END].

---

### Judge Template for CARL and Evo-Verify Data Filtering

You are a Meta-Judge assistant. Your task is to evaluate the quality and correctness of a '[JUDGE LLM RESPONSE]' based on how well it fulfilled the '[REQUIREMENT]' prompt.

You will be given three pieces of information:

1. '[REQUIREMENT]': The specific instructions given to the Judge LLM. 2. '[GROUND TRUTH]': The ground truth answer for the question in '[REQUIREMENT]', not the answer of the '[REQUIREMENT]', for reference. 3. '[LLM RESPONSE]': The full output from the Judge LLM, which may include its reasoning process (e.g., in " tags) and its final answer.

Please evaluate the '[JUDGE LLM RESPONSE]' based on the following three criteria, providing a score from 0 (completely failed) to 10 (perfectly executed) for each.

**Evaluation Criteria:**

1. **Format & Form Adherence Score: 0-10 Points**

* **10 Points (Perfect):** The **final output section** of the Judge LLM (excluding the reasoning process) is **fully and strictly consistent with** the format required in '[REQUIREMENT]'. This includes, but is not limited to: the required data structure (e.g., JSON, Markdown), key names, data types, punctuation marks, and any restrictions such as "output only XX". * **5 Points (Partial):** The overall structure is correct, but there are some deviations. For example: it includes extra explanatory text that the requirement explicitly prohibits adding, has JSON format errors, mismatched key names or data types, or uses an overall format different from the one required in the requirement. * **0 Points (Failure):** The output format is completely inconsistent with the requirements. Special Note: If the requirement includes reasoning tags such as " and " but they are missing from the final output, directly assign a score of 0 for the Format & Form Adherence Score.

2. **Overall Content Correctness Score: 0-10 Points**

* **10 Points (Perfect):** The **entire analysis process and conclusions** of the Judge LLM are completely correct. It accurately understands all the details of '[REQUIREMENT]' and applies them flawlessly to the evaluation of '[INPUT DATA]', with impeccable reasoning logic. * **5 Points (Partial):** The final conclusion may be correct, but there are obvious flaws in the reasoning process or misunderstandings of some minor details in '[REQUIREMENT]'. Alternatively, the reasoning process is generally correct, but there are minor deviations in the final conclusion. * **0 Points (Failure):** The entire analysis and conclusions are based on a wrong understanding, and the task specified in '[REQUIREMENT]' is not completed at all.

3. **Key Conclusion Correctness Score: 0-10 Points**

* **10 Points (Perfect):** The **most core, critical, and important evaluation conclusions** of the Judge LLM are completely correct. The key analysis results of the Judge LLM are logically sound, and the final scores must be reasonable. This criterion does not consider minor flaws in the reasoning process and only focuses on the final decisive judgment. * **5 Points (Partial):** The most core evaluation conclusions are correct, but there are some minor flaws in the reasoning process or misunderstandings of some minor details in '[REQUIRE-

MENT]'. * **0 Points (Failure):** The most core evaluation conclusions are completely incorrect.

Note that the above scores all range from 0 to 10 (integers only).

**Output Requirements:** Format your final evaluation result as a JSON object in the end of your judgement. Ensure the JSON object has the correct format and includes all fields as required, i.e., the four fields: format_score, content_score, key_conclusion_score, and reason.

**JSON Output Format Example:**

```json
{ "format_score": 0, "content_score": 5, "key_conclusion_score": 3, "reason": "..." }
```

——-

**Now, please evaluate based on the following information:**

**[REQUIREMENT]:** require content

requirement

**[GROUND TRUTH]:**

ground_truth

**[LLM RESPONSE]:**

llm_response

Please evaluate the [LLM RESPONSE] based on [REQUIREMENT] and [GROUND TRUTH] in accordance with the above rules:

## A.6 USE OF LLMs

We use LLMs (Gemini-2.5-pro-thinking (Team et al., 2024) to polish writing, find the typos and make the writing more native.

