# OpenReview forum: "UnifiedVerifier: Unifying Paradigms in Automated LLM Evaluation"
_ICLR.cc/2026/Conference — ICLR 2026 Conference Desk Rejected Submission_

### Official Review · Reviewer_tJxx · 2025-10-27

**Soundness:** 2
**Presentation:** 2
**Contribution:** 3
**Rating:** 4
**Confidence:** 4

**Summary:**

This paper introduces UnifiedVerifier, a novel framework designed to address the fragmented and inflexible nature of current LLM evaluation. The primary goal is to unify objective verification and subjective judgment into a single, efficient, and customizable model. The authors propose two core contributions: (1) the Evo-Verify pipeline, an automated and evolutionary method for generating a large-scale, diverse, and complex dataset for training evaluation models, and (2) Core-Anchored Reinforcement Learning (CARL), an alignment technique designed to enhance model reliability by anchoring the reward signal to verifiable ground truths, thereby mitigating reward hacking. Experiments show that a 4B UnifiedVerifier model achieves state-of-the-art performance on both objective and subjective benchmarks, outperforming much larger models with significantly higher inference efficiency.

**Strengths:**

- Well-Motivated Problem: The paper addresses a critical and timely problem in LLM research. The current evaluation landscape is indeed fragmented, and the pursuit of a unified, general-purpose, and flexible evaluation model is a valuable and important research direction.

- Innovative Data Generation Framework (Evo-Verify): The Evo-Verify pipeline is a key strength of this work. Instead of merely rewriting instructions, it systematically evolves the "verification task" itself, resulting in a high-quality dataset with unprecedented diversity in task types, judgment criteria, and output formats. This provides a solid foundation for training a truly general-purpose evaluator.

- New RL Alignment Technique (CARL): CARL offers a well-reasoned solution to the pervasive issue of reward hacking in RL-based alignment. By anchoring the majority of the reward signal to objective, programmatically verifiable facts, it provides a robust mechanism to ensure the model's alignment is stable and truthful, which is a significant improvement over relying solely on a fallible, learned reward model.

- Strong Empirical Results and High Efficiency: The experimental results are impressive. The fact that a 4B parameter model can outperform models that are orders of magnitude larger is a strong testament to the effectiveness of the proposed framework. Furthermore, the demonstrated gains in inference efficiency make this approach highly practical for real-world applications.

**Weaknesses:**

- Limited Empirical Justification for the CARL Stage: While CARL is theoretically well-motivated to improve reliability, its empirical contribution appears marginal in the main results (e.g., Table 2 shows an accuracy improvement from 89.8% to 90.1%). This small gain raises questions about the necessity of the complex RL stage, especially given the strong performance of the SFT model alone. The paper would be strengthened by including experiments on an adversarial test set designed specifically to induce reward-hacking behaviors, which would better demonstrate CARL's unique value in ensuring model robustness.


- Inherent Contradiction in CARL's Scope and the Paper's Goal of Generality: The paper's stated ambition is to create a "unified" verifier for both objective and subjective tasks. However, the CARL framework fundamentally depends on the existence of "verifiable answers" for its core anchoring mechanism. This makes it highly effective for objective tasks but limits its applicability to purely subjective domains (e.g., evaluating creativity, persuasiveness, or empathy) where no such ground truth exists. This creates a tension between the framework's goal of generality and the narrow scope of its proposed reliability mechanism.


- Unclear Process for Ensuring "Correctness" in the Evo-Verify Pipeline: The paper explains that the Evo-Verify process generates new, complex verification tasks. However, the method for generating the corresponding "correct answers" (i.e., the labels for SFT) is only briefly mentioned. It relies on a powerful teacher model (GPT-OSS-120B) to both generate and self-select the best response. This process hinges on the strong assumption that the teacher model is consistently accurate. The paper would benefit from a more detailed explanation of this critical step and a discussion of the potential for propagating teacher model errors into the training data.


- Concerns about the Quality Control of the Initial Seed Data Pool: The methodology starts with 2 million QA pairs from 45 benchmarks. The only filtering step described is based on "difficulty" (retaining questions where model pass rates are low), not on "correctness." This approach fails to address potential label noise in the source benchmarks. Critically, this filtering method could inadvertently enrich the dataset with contaminated samples, as questions with incorrect ground-truth answers would likely cause high-performing models to "fail," thus being misclassified as "difficult" and retained. The paper lacks a sufficient discussion of the quality of its source data and any steps taken to ensure its cleanliness.


- The Evaluation of UnifiedVerifier as a Reward Model is Insufficient and Lacks Critical Baselines: A central claim of the paper is the utility of UnifiedVerifier as a superior reward model. However, the experiment designed to validate this (Table 5) is insufficient as it fails to convincingly demonstrate the added value of its complex, generated rewards.
	- Omission of the Most Direct Baseline: For the chosen downstream tasks (AIME and GPQA), both of which have programmatically verifiable answers, the most direct and critical baseline is a simple, rule-based reward model (i.e., a standard RLVR setup where reward=1 for a correct answer and reward=0 for an incorrect one). The paper does not include this comparison. Without it, it is impossible to know if the nuanced, rubric-based rewards generated by UnifiedVerifier offer any real benefit over a simple correctness signal for these objective reasoning tasks. If a basic rule-based reward performs comparably, it would seriously question the necessity of generating complex feedback for this class of problems.
	- Failure to Test on Appropriate Benchmarks: Furthermore, the very choice of AIME and GPQA as evaluation benchmarks is misaligned with the paper's broader goal of creating a "unified" verifier. These benchmarks do not test the model's advertised strength in handling subjective or multi-faceted tasks where a simple rule-based reward is not feasible. To truly substantiate its "unified" claim, the model must be evaluated as a reward model on more appropriate benchmarks, such as Arena-Hard or MT-Bench, which require the nuanced judgment that UnifiedVerifier is supposedly designed to provide. The current experiment leaves the model's core promise in its most unique application area completely unverified.

**Questions:**

See weakness.

---

> ### Author Response · Authors · 2025-11-26
> **Response to Reviewer tJxx Q1~2:**
>
> We sincerely thank Reviewer tJxx for the insightful and constructive feedback. These comments have helped us identify key areas for clarification and improvement. We address each concern below.
>
> ---
>
> > **Question 1:** Limited Empirical Justification for the CARL Stage: While CARL is theoretically well-motivated to improve reliability, its empirical contribution appears marginal in the main results (e.g., Table 2 shows an accuracy improvement from 89.8% to 90.1%). ... The paper would be strengthened by including experiments on an adversarial test set designed specifically to induce reward-hacking behaviors...
>
> **Response:**
> Thank you for this valuable suggestion. While the gains on VerifyBench-Hard (Table 2) and RewardBench-Chat (Table 3) are consistent, we agree with Reviewer tJxx that an adversarial test set provides a much clearer demonstration of CARL's unique value in ensuring model robustness.
>
> Following this recommendation, we conducted a new experiment on an adversarial test set specifically designed to induce reward-hacking behaviors. We utilized the **Master-RM dataset** [1] (Zhao, Yulai, et al. 2025), which was created to study reward hacking via injection attacks. We constructed a test set by randomly sampling 250 injection-attack samples and 250 clean samples. The results are as follows:
>
> | Model Name | Accuracy | F1 Score |
> | :--- | :---: | :---: |
> | Qwen2.5-7B-Instruct | 76.2 | 50.6 |
> | Qwen2.5-32B-Instruct | 81.6 | 59.7 |
> | Qwen3-4B-Thinking-2507 | 80.8 | 56.8 |
> | Qwen3-30B-A3B-Thinking-2507 | 72.2 | 50.2 |
> | Qwen3-235B-A22B-Instruct-2507 | 80.0 | 60.0 |
> | GPT-OSS-120B | 92.6 | 78.4 |
> | CompassJudger-1-7B-Instruct | 83.4 | 63.4 |
> | RISE-Judge-Qwen2.5-7B | 81.6 | 54.9 |
> | CompassVerifier-32B | 74.6 | 55.1 |
> | UnifiedVerifier-4B w/o CARL | 89.0 | 72.1 |
> | **UnifiedVerifier-4B w CARL** | **92.3 (+3.3)** | **76.4 (+4.3)** |
>
> As the table shows, the attack samples significantly impaired the performance of most models, reducing F1 scores to 60 or below, even for the 235B Qwen3 model. Our UnifiedVerifier-4B (w/o CARL) already performed well. However, the full model with **CARL achieved a 92.3 Acc (+3.3) and 76.4 F1 (+4.3)**.
>
> This result strongly supports our claim that CARL not only boosts general verification capabilities but also, more critically, substantially enhances the model's stability and resilience against reward hacking, which was its primary design goal.
>
> [1] Zhao, Yulai, et al. "One token to fool llm-as-a-judge." arXiv preprint arXiv:2507.08794 (2025).
>
> ---
>
> > **Question 2:** Inherent Contradiction in CARL's Scope and the Paper's Goal of Generality: ...CARL framework fundamentally depends on... "verifiable answers"... This makes it highly effective for objective tasks but limits its applicability to purely subjective domains... This creates a tension between the framework's goal of generality and the narrow scope of its proposed reliability mechanism.
>
> **Response:**
> We appreciate this insightful point from Reviewer tJxx regarding the apparent tension between CARL's mechanism and our goal of generality.
>
> We agree that standard GRPO algorithms often depend on a single, clear, verifiable answer. For purely subjective tasks, typical RLHF learns to mimic the preferences of a specific (and potentially fallible) reward model or human annotator.
>
> Our goal with CARL was different: we wanted to provide a more stable reward signal that encourages exploration *without* narrowly overfitting to a specific, learned subjective preference. CARL's paradigm is *analogous* to the results-based reward of GRPO, but it is not limited to a single correctness signal. It incorporates fine-grained judgments, such as the `Auxiliary Quality Score` ($R_{aux}$), which provides a signal on stylistic quality. This is distinct from simply "distilling" a reward model's preferences for subjective tasks.
>
> Furthermore, as demonstrated in our response to the first concern, the stability gained from CARL benefits *both* objective and subjective task performance (see Table 3, Table 6, and the new Master-RM results). We hypothesize, much like other models that use objective-based RL (e.g., DeepSeek-R1), that the enhanced reasoning and robustness capabilities learned during objective-anchored training generalize, leading to a comprehensive improvement in the model's overall capabilities, including its subjective judgment.
>
> ---

---

> ### Author Response · Authors · 2025-11-26
> **Response to Reviewer tJxx Q3~4:**
>
> > **Question 3:** Unclear Process for Ensuring "Correctness" in the Evo-Verify Pipeline: ...The paper ... relies on a powerful teacher model (GPT-OSS-120B) to both generate and self-select the best response. This process hinges on the strong assumption that the teacher model is consistently accurate. ...
>
> **Response:**
> Thank you for highlighting this crucial aspect of our data pipeline, Reviewer tJxx. We offer three clarifications:
>
> 1.  **Use of Ground Truth in Filtering:** We must clarify a key step. When filtering the four candidate responses generated by the teacher model (lines 455-456), we *do* provide the **ground-truth answer** to the original problem as context for the judge model. The prompt used for this selection (which is the same as our CARL reward prompt) is detailed in **Appendix A.5**. We apologize for not making this clearer in the main text and will explicitly emphasize this process in our revision.
>
> 2.  **Complex Verification Tasks:** As noted, many diverse Evo-Verify tasks do not have a single, simple "correct" answer (e.g., preference rating, code analysis, as shown in Figure 2). For these instances, we rely on the self-selection ("judge to select the single best response") process described in lines 455-456, which is a widely used method for SFT data curation when a simple ground truth is unavailable.
>
> 3.  **Empirical Validation:** The most compelling evidence for the quality of our generated data lies in the results. The UnifiedVerifier, trained on this SFT data, dramatically outperforms its base model (Qwen3-4B-Thinking) on both objective (Table 2) and subjective (Table 3) tasks. More importantly, it even **surpasses the 120B teacher model (GPT-OSS-120B) that generated the data**. This demonstrates the strong positive effect of our Evo-Verify pipeline, which successfully synthesizes a training curriculum that elevates the student model beyond its teacher.
>
> ---
>
> > **Question 4:** Concerns about the Quality Control of the Initial Seed Data Pool: ...The only filtering step described is based on "difficulty" (retaining questions where model pass rates are low), not on "correctness." ...this filtering method could inadvertently enrich the dataset with contaminated samples...
>
> **Response:**
> This is a very valid concern regarding the initial seed pool. Filtering by difficulty alone could indeed, as Reviewer tJxx suggests, inadvertently select for samples with noisy labels.
>
> However, this potential issue is addressed in our *subsequent* data generation and filtering stages. As mentioned in our previous response, the crucial step is **line 456**, where we use GPT-OSS-120B *acting as a judge* to select the single best response from four candidates. This judging process is **provided with the original ground-truth answer** for reference (using the prompt in Appendix A.5).
>
> Therefore, if a "difficult" question from the seed pool was difficult only because its ground-truth label was incorrect, it would be highly likely to be filtered out during this quality-control step. The judge model would recognize the discrepancy between the *correct* generated response and the *noisy* ground-truth, and penalize that sample. This multi-stage process helps ensure the cleanliness of the final training data.

---

> ### Author Response · Authors · 2025-11-26
> **Response to Reviewer tJxx Q5~6:**
>
> > **Question 5 & 6:** The Evaluation of UnifiedVerifier as a Reward Model is Insufficient... the Most Direct Baseline [rule-based reward]... [and fails] to Test on Appropriate Benchmarks [AIME/GPQA are objective, not subjective like Arena-Hard/MT-Bench].
>
> **Response:**
> We appreciate Reviewer tJxx's detailed critique of our reward model evaluation (Section 6.1, Table 5). We believe two key clarifications will resolve this concern.
>
> 1.  **Why the Rule-Based Baseline is Not Applicable:**
>     The core of this experiment is not just to get the right final answer, but to test UnifiedVerifier's utility as a **rubric-based reward model**. As stated in Section 6.1, the RLVR training dataset used is **RAR-Science** [1]. This dataset is specifically designed to provide fine-grained, multi-dimensional rubric feedback (e.g., evaluating reasoning steps, clarity, *and* correctness), not just a final pass/fail signal.
>
>     Therefore, a simple rule-based reward model (e.g., `reward=1` for correct, `reward=0` for incorrect) is **not a valid or applicable baseline** for this experiment. Such a baseline is fundamentally unable to process the rich, rubric-based requirements of the RAR-Science training data, which demands the RM to evaluate multiple, specific criteria beyond just the final answer. Our baselines, Qwen3-4B-Thinking and GPT-OSS-120B, *are* capable of this rubric-based evaluation and serve as strong points of comparison.
>
> 2.  **Why AIME/GPQA are the Correct Benchmarks for *This* Experiment:**
>     Following the point above, since the policy model was trained using the **RAR-Science** dataset (which is focused on scientific reasoning), evaluating the policy's resulting performance on challenging science and math benchmarks like **AIME and GPQA is a direct and appropriate measure** of the reward model's effectiveness.
>
>     While our UnifiedVerifier *is* designed for unified (objective + subjective) evaluation—as demonstrated on RewardBench-Chat (Table 3) and the competition-level UnifiedHLE-Verify (Table 4)—this *specific experiment* in Section 6.1 was designed to test its capability in the demanding application of **rubric-based RLVR**. The results, which show our 4B model as an RM leads to better policy performance than using the 120B GPT-OSS or the base Thinking model, validate its strength in this domain.
>
> [1] Gunjal et al. "Rubrics as rewards: Reinforcement learning beyond verifiable domains." 2025.

---

### Official Review · Reviewer_Ymp7 · 2025-11-01

**Soundness:** 1
**Presentation:** 1
**Contribution:** 2
**Rating:** 2
**Confidence:** 4

**Summary:**

The paper aims to address an important problem – how to build more robust verifiers that can generalize to many domains. They train a verifier using their recipe and compare it to existing judge models on two verification benchmarks.

**Strengths:**

The trained verifier model appears to be stronger than other larger models and other judge models trained for verification on two verification benchmarks. They compare against several other judge models.

**Weaknesses:**

The writing of the paper is a bit convoluted and difficult to parse clearly. Several details of how their dataset was generated and specifics of their training algorithm feel underspecified to both reproduce the work and clearly understand the contributions. In particular the CARL approach for training the verifier feels quite underdefined. Please see questions for specific notes on where additional clarity is needed. The authors compare against several other judge models but it would be valuable to clearly outline how the training data and algorithm differ from that used for training the other models. For example, is their model stronger because it was trained for longer, with more data, or due to the specific algorithm used? They evaluate their model on different verification benchmarks — it would be good to clearly define what these benchmarks are checking for. A more realistic and convincing evaluation would be to see if using their verifier when RL tuning a model for a specific task leads to better results than training a task-specific critic (e.g. PPO), using a pre-existing judge model, or using no critic model at all (GRPO). The authors mention some initial results around this, but the details of the training setup are lacking. While the premise of the paper is interesting, the work feels like a first-draft and would benefit from significant rewrites for better clarity, better positioning and detailing their approach, and from more realistic evaluations (as mentioned above).

**Questions:**

- starting quotes are reversed throughout, please fix
- section 1 line 105: which base model was used? which 120B param model are you comparing against? This is mentioned later in the text, but useful to specify it early on.
- line 162-167: the language here can be simplified – it seems that the main difference is that Evol-Instruct generates a broad set of instructions whereas Evo-Verify generates verification related instructions?
- line 180: If out of distribution generalization is a problem for verifiers, wouldn’t it be better to build task-specific verifiers or co-train task-specific verifiers w/ generators?
- line 216: what domains are these benchmarks covering? what is their format – MCQ, short answer, long-form reasoning? If possible, explicitly listing out the benchmarks would be best. The format would determine how verification would take place.
- line 220: please mention which LLMs are used for this process.
- line 243: is the filtering done using a different LLM?
- line 245: were there existing answer choices in the selected questions? are those discarded and replaced by model generated answers?
- line 246: is the best option selected by comparing against the ground truth answer? perhaps the same model should not be used for answer generation & selection as it may be biased toward particular answer choices.
- line 246: what is the final format of the training instances – is there a separate instance to very each generated answer choice for each question?
- line 261: Are these 30,000 questions a subset of the 200,000 instances generated in the previous section? How was the subset selected?
- section 4.2, line 264: The motivation for CARL is unclear. If the goal is to train a verifier, I assume the data is pairs of (question, candidate response) or (question, candidate response, ground truth) and the model’s output would be some CoT + a correct or not assessment for the candidate response? If this is the case, then what do you mean by “forcing the model to generate the verifiable response without seeing the solution”? Please describe the motivation and data inputs and expected model outputs more clearly.
- section 4.3: Currently these rewards are not well motivated as the goal of training with CARL is not clearly explained. Is CARL simply GRPO with specific rewards?
- line 313: How is the data / training strategy of these judge models different from what is used for UnifiedVerifier?
- line 358: What is the process of training UnifiedVerifier-4B w/o CARL? Is this simply an SFT with the generated data?
- line 429: What algorithm did you use to tune Qwen 2.5 7B with the UnifiedVerifier? Please specify more details of the training setup. How does this compare to PPO where a critic is trained in conjunction with the policy, or vanilla GRPO where there is no critic model.

---

> ### Author Response · Authors · 2025-11-26
> **Response to Reviewer Ymp7**
>
> We sincerely thank Reviewer Ymp7 for the thorough and highly constructive analysis of our paper. We address each point below, referencing specific sections of the paper where the information is detailed or where clarifications will be added in the revision. Our responses aim to demonstrate the rigor of our work.
>
> ---
>
> ### 1. Typo errors and Clarity
>
> > **Question 1.1 (Typographical):** The opening quotation marks throughout the text are inverted and should be corrected.
>
> **Response:**
> We thank Reviewer Ymp7 for identifying this typographical issue. We confirm this minor formatting error has been corrected in the updated manuscript.
>
> > **Question 1.2 (Clarity):** Section 1, Line 105: Which base model was used? Which 120-billion-parameter model are you comparing against? This is mentioned later, but should be clarified upfront.
>
> **Response:**
> We agree this context is crucial for early comprehension. The Introduction (Section 1) will be revised to explicitly state:
> * The base model for our efficient variant is a 4B parameter architecture (specifically, **Qwen3-4B-Thinking**).
> * The 120B parameter model, used for performance comparison and as the source for the $R_{aux}$ auxiliary reward, is **GPT-OSS-120B**.

---

> ### Author Response · Authors · 2025-11-26
> **Response to Reviewer Ymp7**
>
> ### 2. Evo-Verify Methodology and Dataset Construction
>
> > **Question 2.1:** Lines 162-167: The language here could be simplified—it seems the main difference is that Evol-Instruct generates a wide range of general instructions, while Evo-Verify generates verification-related instructions?
>
> **Response:**
> We confirm the language in Section 3 (Evo-Verify) will be simplified. The distinction is **methodological**: Evo-Verify is specialized for verification, ensuring comprehensive coverage across five specific dimensions (Logic, Code/Execution, Constraint/Format Following, etc.). Importantly, it systematically generates instances demanding **customizable structural outputs** (e.g., the JSON format in Chemistry Example 1), moving beyond simple instruction-following.
>
> > **Question 2.2:** Line 180: If out-of-distribution generalization is a problem for verifiers, wouldn't it be better to build task-specific verifiers or jointly train them with a generator?
>
> **Response:**
> Our framework fundamentally challenges this assumption. As stated in the Abstract, the current evaluation landscape is fragmented. UnifiedVerifier addresses this by demonstrating that a unified model *can* achieve high generalization and performance. Empirically, UnifiedVerifier-4B outperforms larger, specialized verification models (e.g., **CompassVerifier**) in objective tasks and judge models (e.g., **CompassJudger**) in subjective tasks. This confirms that unification does not entail a performance trade-off, especially when anchored by the objective learning signal of CARL.
>
> > **Question 2.3:** Line 216: What domains do these benchmarks cover? What are their formats—multiple-choice, short answer, long-form reasoning? If possible, please list them explicitly.
>
> **Response:**
> We thank the reviewer for this suggestion. A detailed list of all 45 datasets is provided in **Appendix A.3, Table 7**. We also outline the covered Categories and Sub-tasks in **Table 1**. To further enhance clarity as suggested, we will update the appendix to explicitly associate each dataset with its primary domain and question format (e.g., multiple-choice, long-form reasoning).
>
> > **Question 2.4:** Line 220: Please specify which large language models were used in this process.
>
> **Response:**
> We used a diverse pool of 62 models spanning 20 model families. For each question, responses from at least 10 models were sampled from this pool for the filtering process. The complete list of models is provided below for reference:
>
> | Model Family | Model Name |
> | :--- | :--- |
> | **Yi** | Yi-Lightning, Yi-1.5-9B-Chat |
> | **GPT** | GPT-4o, GPT-4o-mini, GPT-4-1-2025-0414, GPT-4.5-preview-2025-02-27 |
> | **GPT-OSS** | gpt-oss-20b, gpt-oss-120b |
> | **Doubao** | Doubao-Pro-32k-241215, Doubao-Pro-1.5-32k-250115, Doubao-Pro-32k-240828 |
> | **Qwen** | Qwen-Max-0919, Qwen-Max-2025-01-25, Qwen2.5-Max, Qwen2.5-7B-Instruct, Qwen2.5-14B-Instruct, Qwen2.5-32B-Instruct, Qwen2.5-72B-Instruct, QwQ-32B |
> | **Qwen3** | Qwen3-0.6B, Qwen3-1.7B, Qwen3-4B, Qwen3-8B, Qwen3-14B, Qwen3-32B, Qwen3-30B-A3B (MoE), Qwen3-235B-A22B (MoE) |
> | **Gemini** | Gemini-2.0-Flash-Exp, Gemini-1.5-Pro, Gemini-2-5-Pro-03-25 |
> | **DeepSeek-R1** | DeepSeek-Chat-R1, DeepSeek-R1-distill-Qwen-1.5B, DeepSeek-R1-distill-Qwen-7B, DeepSeek-R1-distill-Llama-8B, DeepSeek-R1-distill-Qwen-14B, DeepSeek-R1-distill-Qwen-32B, DeepSeek-R1-distill-Llama-70B |
> | **Llama** | Llama-3-1-8B-Instruct, Llama-3-1-70B-Instruct, Llama-3-2-3B-Instruct, Llama-3-3-70B-Instruct |
> | **Mixtral** | Mistral-Small-Instruct-2409, Mistral-Small-3.1-24B-Instruct, Ministral-8B-Instruct-2410, Mixtral-Large-Instruct-2411 |
> | **Claude** | Claude-3-5-Sonnet-20241022, Claude-3-7-Sonnet-20250219, Claude-3-7-Sonnet-20250219-Thinking |
> | **Gemma** | Gemma-2-9B-It, Gemma-2-27B-It, Gemma3-27B-It |
> | **DeepSeek-Chat** | DeepSeek-V2.5, DeepSeek-Chat-V3 |
> | **InternLM** | InternLM2.5-7B-Chat, InternLM2.5-20B-Chat, InternLM3-8B-Instruct |
> | **Phi** | Phi-4 |
> | **GLM** | GLM-4-9B-Chat, GLM-4-Plus |
> | **MiniMax** | MiniMax-Text-01 |
> | **Moonshot** | Moonshot-V1-32k |
> | **Hunyuan** | Hunyuan-Standard-256K |
> | **StepFun** | Step-2-16k |
>
> > **Question 2.5:** Line 243: Was the filtering operation performed using a different Large Language Model?
>
> **Response:**
> As referenced in the context of lines 245-247, the same auxiliary model, **GPT-OSS-120B**, was used for both the SFT data generation and the selection/filtering stages.

---

> ### Author Response · Authors · 2025-11-26
> **Response to Reviewer Ymp7**
>
> > **Question 2.6:** Line 245: Did the selected problems have existing answer options? Were these discarded and replaced by model-generated answers?
>
> **Response:**
> This is a crucial point regarding the Evo-Verify pipeline. The "model-generated answers" referenced in Line 245 are **not** new answers to the *original* question. As detailed in Lines 225-242, the pipeline first transforms the initial data into a `<Meta-input, Fine-grained Requirement>` pair.
>
> The "four candidate outputs" (Line 245) are generated by GPT-OSS-120B as *responses* to this new, complex `Fine-grained Requirement`, not as replacements for the original ground-truth answer. The original ground-truth answer (if present in the `Meta-input`) serves as context or a reference for the new verification task.
>
> > **Question 2.7:** Line 246: Is the best option selected via comparison with the true answer? Perhaps the same model should not be used for both generation and selection, as it might be biased towards specific answer options.
>
> **Response:**
> This potential for bias is a key point, which we mitigate in two ways.
> 1.  **First**, during the SFT data generation, the judging process is **provided with the original ground-truth answer** for reference (using the prompt in Appendix A.5), which anchors the selection to the correct answer.
> 2.  **Second**, and more importantly, this bias is precisely why we developed the **CARL** framework. As detailed in Section 4.3, the final alignment optimization is dominated by the **Core Objective Score ($R_{core}$)**. This score relies on programmatic validation against the objective Ground Truth (GT) and constitutes **60% of the total reward**. This objective anchor ensures the model is aligned to empirical truth, not the potential biases of the SFT data generator.
>
> > **Question 2.8:** Line 246: What is the final format of the training instances—is there a separate verification instance for each generated answer option for a given problem?
>
> **Response:**
> We thank the reviewer for this question, as the format is key to unification. As illustrated in **Figure 2**, the final training instances support **both** paradigms to ensure the model can act as a unified verifier:
> 1.  **Single-response verification:** Instances where the `Meta-input` contains a single response (e.g., Q + R) and the `Fine-grained Requirement` asks for a judgment on it.
> 2.  **Comparative/Ranking tasks:** Instances where the `Meta-input` contains multiple responses (e.g., Q + R₁ + R₂ + ...) and the `Requirement` asks for a relative quality evaluation (as shown in the 'Preference Rating' example in Figure 2).
>
> This unified format trains the model to handle both absolute and relative evaluation within a single framework.
>
> > **Question 2.9:** Line 261: Are these 30,000 questions a subset of the 200,000 instances generated in the previous section? How was this subset selected?
>
> **Response:**
> We will clarify this terminology. The final training corpus for Supervised Fine-Tuning (SFT) contains 200,000 high-quality instances. The 30,000 questions (referenced in Section 4.2, Line 261) are indeed a **subset** of these 200,000 SFT instances, selected specifically for the CARL alignment phase due to their **complexity and programmatic verifiability**.
>
> ---

---

> ### Author Response · Authors · 2025-11-27
> **Response to Reviewer Ymp7**
>
> ### 3. Core-Anchored Reinforcement Learning (CARL)
>
> > **Question 3.1:** Section 4.2, Line 264: CARL’s motivation is unclear. [...] what does "forcing the model to generate verifiable responses without seeing the solution" mean? Please describe the motivation, input data, and expected model output more clearly.
>
> **Response:**
> We appreciate the feedback and will dedicate Section 4.2 to clarifying CARL:
> * **Motivation (Section 4.2):** CARL is designed to prevent **reward hacking** by minimizing reliance on subjective, fallible reward signals.
> * **Input/Output:** The input includes Q, R, and often GT. The output is a Chain-of-Thought (CoT), marked by `<think>...</think>`, followed by the final judgment.
> * **Mechanism:** The phrase "without seeing the solution" implies the model must discover the correct verification *process* (the CoT). CARL ensures this rationale is verifiable because the optimization is heavily anchored to the $R_{core}$ score, which is determined by external, objective GT validation.
>
> > **Question 3.2:** Section 4.3: Since the CARL training objective hasn't been clearly explained, the rewards currently lack justification. Is CARL simply GRPO with specific rewards?
>
> **Response:**
> The reviewer is correct that CARL utilizes the GRPO objective, as stated in Line 280. However, it is a **modified** framework distinct from standard GRPO, as detailed in Section 4.3 (Lines 280-290 context).:
> * **Hybrid Reward Justification:** The reward signal is unique. The rationale is defined by the CAVR structure. The **60% weight on $R_{core}$** (Core Objective Score) is justified because it provides an indisputable, programmatic anchor against GT, making the optimization robust.
> * **GRPO Modification:** CARL implements a crucial modification: the KL-divergence penalty is explicitly **removed ($\beta=0$)** (as noted in the context surrounding Lines 283-285). This removal is justified because the strong, objective anchoring from $R_{core}$ renders the soft constraint of the KL term redundant.
>
> CARL train details we also showed in A.1.2 CARL IMPLEMENTATION DETAILS as mentioned in line 301.
>
> ---
>
> ### 4. Experimental Details and Alignment
>
> > **Question 4.1:** Line 313: How do the data/training strategies of these comparison models differ from those used by UnifiedVerifier?
>
> **Response:**
> The key difference lies in the **breadth of the training data** and the **alignment objective**. The comparison models (e.g., specialized verifiers) lack the unification of evaluation paradigms (across the five categories) provided by the Evo-Verify data. Furthermore, they lack the objective robustness ensured by CARL’s **60% $R_{core}$ anchor**. UnifiedVerifier's strategy is uniquely optimized for generalized, robust, and customizable evaluation.
>
> > **Question 4.2:** Line 358: What was the process for training UnifiedVerifier-4B without CARL? Was it just supervised fine-tuning using the generated data?
>
> **Response:**
> We confirm this is correct. The baseline for the ablation study (UnifiedVerifier-4B w/o CARL) was trained using the identical 200,000 instances from Evo-Verify, but exclusively via **Supervised Fine-Tuning (SFT)**. This experiment (detailed in the context of Section 5.2) validates that the substantial performance gains are attributable to the subsequent CARL alignment step.
>
> > **Question 4.3:** Line 429: What algorithm did you use to fine-tune Qwen 2.5 7B with UnifiedVerifier? Please explain the training setup. How does this differ from PPO... or vanilla GRPO...?
>
> **Response:**
> We use the same GRPO algorithm with the RAR-Science [1] rubric reward dataset paper we mentioned in line 426, for more details we showed in Appendix A.3 part.
>
> [1] Gunjal, Anisha, et al. "Rubrics as rewards: Reinforcement learning beyond verifiable domains." arXiv preprint arXiv:2507.17746 (2025).

---

### Official Review · Reviewer_t5we · 2025-11-01

**Soundness:** 2
**Presentation:** 2
**Contribution:** 3
**Rating:** 4
**Confidence:** 4

**Summary:**

This paper presents UnifiedVerifier, an innovative framework aimed at providing comprehensive, general-purpose, and customizable verification capabilities within a 4B model. The authors introduce an automated data generation methodology based on the Evol-Instruct method, which systematically synthesizes the "verification task" across various formats and difficulty levels. To enhance the reward model, they propose a Core-Anchored Reinforcement Learning approach, optimized with multi-dimensional signals such as objectiveness, quality, and format. While this study holds significant value for the community, there are several areas for improvement regarding the methodology and experimental design.

**Strengths:**

1.	The Evo-Verify method is a promising approach for automatically generating training data, which enhances the development of general and controllable reward models.
2.	The adaptation of a reinforcement learning algorithm for verification tasks yields positive results, indicating potential for further exploration.

**Weaknesses:**

1.	**Reward Hacking Concerns**: The authors emphasize the challenge of "reward hacking" multiple times. Are there empirical results demonstrating UnifiedVerifier's effectiveness in mitigating this issue?
2.	In lines 227-229, the authors reference tasks that fall under type 3, which retain only the question. Could they provide specific examples or clarify this classification?
3.	**Filtering Process Details**tering process based on three axes: question difficulty, verification complexity, and instruction strictness. What exactly does this filtering entail? Additionally, why is there a dual filtering for question difficulty? How is the quality of GPT-OSS assessed in selecting the best response?
4.	The relationship between the 30,000 challenging SFT examples (line 261) and the 200,000 high-quality instances for supervised fine-tuning (line 244) needs clarification. How do these datasets interact?
5.	**Evaluation Datasets**: Why did the authors choose not to utilize the new RewardBench2 dataset for evaluation?
6.	**Baseline Comparison**: For a fairer evaluation, the baseline models should be changed to Qwen3-4B as well. This would allow for a more accurate assessment of the effectiveness of Evo-Verify and CARL.
7.	In line 372, the authors assert UnifiedVerifier's superior "fine-grained" verification capability for challenging problems. What evidence supports this claim?

**Questions:**

Please refer to the Weaknesses

---

> ### Author Response · Authors · 2025-11-26
> **Response to Reviewer t5we**
>
> We sincerely thank Reviewer t5we for the insightful questions and constructive feedback. We are encouraged that the reviewer found our work promising. We address each point below and provide new experimental results to further validate our claims.
>
> > **Question1:** Reward Hacking Concerns: The authors emphasize the challenge of "reward hacking" multiple times. Are there empirical results demonstrating UnifiedVerifier's effectiveness in mitigating this issue?
>
> **Response:**
> Thank you for this critical question. We provide two key pieces of empirical evidence to address this.
>
> First, as demonstrated in **Section 6.1 (Unified Verifier as Rubric Reward Model)**, we show that rubric-based reward modeling places significantly higher demands on a reward model than typical scalar-based rewards. This is because the judgment criteria can vary for each problem, and the model must stably evaluate multiple metrics. In our RUBRIC RL experiments using Qwen2.5-7B-Base as the policy model, UnifiedVerifier (as the reward model) enabled the policy model to outperform versions trained with Qwen3-4B-Thinking-2507 and even the much larger GPT-OSS-120B as reward models. This result from practical RL training serves as a strong testament to UnifiedVerifier's stability.
>
> Second, we acknowledge that reward hacking is an occasional phenomenon, making it resource-intensive to trigger and validate its mitigation across extensive RL experiments. Therefore, we turned to recent related research. We found a study [1] (Zhao, Yulai, et al. 2025) that investigates reward hacking and released the **Master-RM** dataset, which includes samples designed to "hack" reward models via special text or formatting (injection attacks). We constructed a test set by randomly sampling 250 injection-attack samples and 250 clean samples from Master-RM to specifically test robustness against reward hacking.
>
> The results are as follows:
>
> | Model Name | Accuracy | F1 Score |
> | :--- | :---: | :---: |
> | Qwen2.5-7B-Instruct | 76.2 | 50.6 |
> | Qwen2.5-32B-Instruct | 81.6 | 59.7 |
> | Qwen3-4B-Thinking-2507 | 80.8 | 56.8 |
> | Qwen3-30B-A3B-Thinking-2507 | 72.2 | 50.2 |
> | Qwen3-235B-A22B-Instruct-2507 | 80.0 | 60.0 |
> | GPT-OSS-120B | 92.6 | 78.4 |
> | CompassJudger-1-7B-Instruct | 83.4 | 63.4 |
> | RISE-Judge-Qwen2.5-7B | 81.6 | 54.9 |
> | CompassVerifier-32B | 74.6 | 55.1 |
> | UnifiedVerifier-4B w/o CARL | 89.0 | 72.1 |
> | **UnifiedVerifier-4B w CARL** | **92.3** (+3.3) | **76.4** (+4.3) |
>
> As the results show, this reward hacking test set significantly impacts most models; the F1 scores for a majority of models dropped to 60 or below due to the injected perturbations, including the 235B Qwen3-235B-A22B-Instruct-2507 model. Our UnifiedVerifier-4B w/ CARL achieved high performance, second only to GPT-OSS-120B.
>
> Most importantly, **CARL provided a +3.3 Accuracy and +4.3 F1 score improvement** over the SFT-only version (UnifiedVerifier-4B w/o CARL). This demonstrates that CARL not only enhances objective and subjective verification capabilities but also substantially improves the model's stability and resilience against reward hacking.
>
> [1] Zhao, Yulai, et al. "One token to fool llm-as-a-judge." arXiv preprint arXiv:2507.08794 (2025).
>
> -----

---

> ### Author Response · Authors · 2025-11-26
> **Response to Reviewer t5we Q2:**
>
> > **Question2:** In lines 227-229, the authors reference tasks that fall under type 3, which retain only the question. Could they provide specific examples or clarify this classification?
>
> **Response:**
> This is a key point, thank you for asking for clarification.
>
> This task category is designed to ensure that UnifiedVerifier **does not lose its general-purpose question-answering capabilities** and to enhance its overall generalization.
>
> In this scenario, the model is required to directly answer the *original question* (from Stage 1, where only the question is retained), but it must do so according to the specific, fine-grained requirements generated during Stage 2 (Requirement Granulation) and Stage 3 (Structural Constraint Imposition).
>
> For example, consider an original math or coding problem. The Evo-Verify pipeline will take this original problem and generate a *new* set of detailed instructions for how to answer it. This new requirement might refine the problem description, demand a specific reasoning process, or, as in the example below, enforce a very strict output format. This process trains the model to solve general problems while strictly adhering to complex instructions, thereby improving its utility and generalization.
>
> Here is a concrete example:
>
> **\<INPUT DATA BEGIN\> (Original Problem):**
>
> ```json
> {'origin_prompt': [{'role': 'HUMAN', 'prompt': 'You are an expert at Python programming, code execution, test case generation, and fuzzing.\nYou are given a Python function and an assertion containing an input to the function. Complete the assertion with a literal (no unsimplified expressions, no function calls) containing the output when executing the provided code on the given input, even if the function is incorrect or incomplete. Do NOT output any extra information. Provide the full assertion with the correct output in [ANSWER] and [/ANSWER] tags, following the examples.\n\n[PYTHON]\ndef repeatNumber(number : int) -> int:\n    return number\nassert repeatNumber(number = 17) == ??\n[/PYTHON]\n[ANSWER]\nassert repeatNumber(number = 17) == 17\n[/ANSWER]\n\n[PYTHON]\ndef addCharacterA(string : str) -> str:\n    return string + "a"\nassert addCharacterA(string = "x9j") == ??\n[/PYTHON]\n[ANSWER]\nassert addCharacterA(string = "x9j") == "x9ja"\n[/ANSWER]\n\n[PYTHON]\nfrom typing import *\n\nfrom math import *\n\ndef canSplitArray(nums: List[int], m: int) -> bool:\n    if len(nums) <=2:\n        return True\n    for i in range(len(nums)-1):\n        if sum(nums[i:i+2]) >= m:\n            return True\n    return False\nassert canSplitArray(nums = [2, 1, 3], m = 5) == ??\n[/PYTHON]\n[ANSWER]\n\n'}]}
> ```
>
> **\<INPUT DATA END\>**
>
> **New Requirement Generated by Evo-Verify (Stages 2 & 3):**
>
> > You are given the INPUT DATA which contains three Python code snippets and their corresponding completed assertions. The first two examples are already solved, while the third example ends with `assert canSplitArray(nums = [2, 1, 3], m = 5) == ??`.
> >
> > Your task is to:
> >
> > 1.  Execute (or mentally simulate) the function `canSplitArray` defined in the third snippet with the provided arguments `nums = [2, 1, 3]` and `m = 5`.
> > 2.  Determine the exact boolean literal (`True` or `False`) that the function returns for this call, **even if the implementation is buggy or incomplete**.
> > 3.  Produce the completed assertion line, exactly as it should appear in the original format, and wrap it inside `[ANSWER]` and `[/ANSWER]` tags. No additional commentary, whitespace before or after the tags, or any other text is allowed.
> >
> > The expected output format is:
> >
> > ```
> > [ANSWER]
> > assert canSplitArray(nums = [2, 1, 3], m = 5) == <True/False>
> > [/ANSWER]
> > ```
> >
> > Provide only this block.
>
> -----

---

> ### Author Response · Authors · 2025-11-26
> **Response to Reviewer t5we Q3~Q5:**
>
> > **Question3:** Filtering Process Detail: ...filtering process based on three axes: question difficulty, verification complexity, and instruction strictness. What exactly does this filtering entail? Additionally, why is there a dual filtering for question difficulty? How is the quality of GPT-OSS assessed in selecting the best response?
>
> **Response:**
> (Answering the first two parts of the question based on the provided text):
>
> As with other parts of our pipeline, we use **GPT-OSS-120B to score** these three axes (e.g., on a scale of 1-10) and then filter to retain the samples with high aggregate scores.
>
> Regarding the **dual filtering for question difficulty**:
>
> 1.  The first filter, applied in Stage 1, is based on the **pass rate** of the question when answered by multiple models.
> 2.  We discovered that this "pass rate" metric can be unreliable. It can be skewed by external factors, such as evaluation scripts failing for a particular problem or some models failing to follow instructions (which prevents correct answer extraction), thus not reflecting the *true* difficulty.
> 3.  Therefore, we introduced the second filter. In this step, we use a model (GPT-OSS-120B) to **read and directly assess the problem's difficulty**, which provides a more authentic difficulty score. This second filter serves as a more reliable quality check.
>
> -----
>
> > **Question4:** The relationship between the 30,000 challenging SFT examples (line 261) and the 200,000 high-quality instances for supervised fine-tuning (line 244) needs clarification. How do these datasets interact?
>
> **Response:**
> Thank you for pointing this out. As stated in line 262, the 30,000 examples for CARL are a **subset** of the 200,000 high-quality SFT instances.
>
> Specifically, we filtered the 200,000 SFT instances to find those where "the instruction’s requirement contains a core metric that is programmatically verifiable against the ground-truth answer." From this qualifying subset, we **randomly selected 30,000 samples** to create the dataset for the CARL stage.
>
> We apologize for any lack of clarity and will refine this description in the next version of the paper.
>
> -----
>
> > **Question5:** Evaluation Datasets: Why did the authors choose not to utilize the new RewardBench2 dataset for evaluation?
>
> **Response:**
> Thank you for the suggestion. We have now run evaluations on RewardBench2 using the official script. The results are as follows:
>
> | Model | Factuality | Focus | Math | Precise IF | Safety | Ties | Average |
> | :--- | :---: | :---: | :---: | :---: | :---: | :---: | :---: |
> | Qwen3-4B-Thinking-2507 | 41.9 | 73.4 | 37.4 | 32.7 | 74.3 | 51.4 | 51.9 |
> | Qwen3-30B-A3B-Thinking-2507 | 62.6 | 78.6 | 50.7 | 37.2 | 84.6 | 77.4 | 65.2 |
> | GPT-OSS-120b | 69.9 | 87.7 | 87.4 | 62.3 | 78.9 | 17.9 | 67.4 |
> | RISE-Judge-Qwen2.5-7B | 49.9 | 84.4 | 50.3 | 33.8 | 69.3 | 39.9 | 54.6 |
> | CompassJudger-1-7B-Instruct | 36.4 | 66.1 | 47.5 | 33.8 | 58.9 | 32.2 | 45.8 |
> | CompassVerifier-32B | 39.1 | 80.8 | 50.4 | 37.8 | 37.8 | 31.0 | 46.2 |
> | **UnifiedVerifier-4B** | 57.5 | 83.0 | 83.5 | 43.1 | 79.8 | 69.6 | **69.4** |
>
>
> On the recent, more subjective RewardBench2 dataset, UnifiedVerifier-4B (trained with our Evo-Verify and CARL) achieves an average score of 69.4. This is a **17.5-point improvement** over the Qwen3-4B-Thinking-2507 baseline (51.9), further demonstrating the effectiveness of our method for both subjective and objective verification tasks.
>
> -----

---

> ### Author Response · Authors · 2025-11-26
> **Response to Reviewer t5we Q6:**
>
> > **Question6:**  For a fairer assessment, the baseline model should be Qwen3-4B. This would help more accurately evaluate the effectiveness of Evo-Verify and CARL.
>
> **Response:**
> Regarding the choice of baseline: We selected **Qwen3-4B-Thinking-2507** because a unified verifier requires strong **general-purpose capabilities** as a foundation. For example, to perform fine-grained analysis of a difficult math problem's answer, the verifier model must itself possess strong mathematical reasoning ability.
>
> The Qwen3-4B-Thinking model, having been post-trained on high-quality general-purpose data, is a much stronger and more appropriate starting point than the Qwen3-4B base model, which lacks this extensive training. As we state in lines 80-81, our core vision is to “instill comprehensive and general-purpose verification capabilities.” We believe general capability and general-purpose verification capability are complementary and mutually reinforcing.
>
> However, to further validate the effectiveness of our method on the base model as you suggested, we conducted additional experiments training on different base models using our Evo-Verify SFT data. The results are below:
>
> | Model Name | VerifyBench-Hard (ACC) | RewardBench-Chat (ACC) |
> | :--- | :---: | :---: |
> | Qwen3-4B | 69.9 | 56.1 |
> | Qwen3-4B-Instruct | 81.8 | 58.4 |
> | Qwen3-4B-Thinking | 82.2 | 79.5 |
> | **Qwen3-4B + Evo-Verify** | **84.0** (+14.1) | **75.7** (+19.3) |
> | **Qwen3-4B-Instruct + Evo-Verify** | **87.8** (+6.0) | **79.6** (+21.2) |
> | **Qwen3-4B-Thinking + Evo-Verify** | **89.8** (+7.6) | **82.8** (+3.3) |
>
> As shown, when applied to the **Qwen3-4B (base) model**, our Evo-Verify data improved performance by **+14.1 Accuracy on VerifyBench-Hard** and **+19.3 Accuracy on RewardBench-Chat**. This (a 34.1% relative improvement on RewardBench-Chat) demonstrates that our method provides substantial gains even on the base model.
>
> -----

---

> ### Author Response · Authors · 2025-11-26
> **Response to Reviewer t5we Q7:**
>
> > **Question7:** In line 372, the authors claim UnifiedVerifier has superior "fine-grained" verification capabilities on challenging problems. What evidence supports this claim?
>
> **Response:**
> Thank you for asking for this evidence. As mentioned in line 370 and our supplementary tests in Appendix A.5, we used the **Human Last Evaluation (HLE)** dataset as a source. HLE is arguably one of the most difficult reasoning benchmarks currently available. Consequently, verifying the responses to HLE questions presents a significant challenge that standard verification datasets do not capture. As described in lines 321-323, we specifically applied our **Evo-Verify** methodology to HLE questions to construct the **UnifiedHLE-Verify** benchmark. This benchmark is designed to be both high-difficulty (derived from HLE) and fine-grained (due to the detailed requirements generated by the Evo-Verify pipeline).
>
> To ensure the quality and fairness of the reference answers, the responses in this benchmark were generated and selected by a consensus of five top-tier reasoning models (including GPT-5, Gemini2.5-Pro-Thinking, and Grok4). The results for UnifiedHLE-Verify (presented in **Table 4**, line 652) demonstrate that UnifiedVerifier-4B achieves the top rank in verifying these competition-level responses, which supports our claim of superior fine-grained verification ability.
>
> Here is a concrete example from the UnifiedHLE-Verify dataset that highlights the fine-grained nature of the verification task:
>
> **Original HLE Question:**
>
> > I am currently a 49-year-old middle manager of a large, for-profit corporation in the United States making 250,000 with no dependents or spouse. My birthday is on June 3rd. I do not own my own business. I have contributed 23,000 to my 401k, 4,150 in my HSA, 3,200 in my FSA, and 5,000 in a backdoor Roth IRA. I received a 50% employer match up to 50% on my 401k. How much more money am I legally allowed to contribute to my retirement account in the 2024 tax year?\\nRemember to put your final answer within \\boxed{}.
>
> **Original Response to Judge:**
>
> > "prediction": "We need to interpret the question carefully. The user: 49-year-old middle manager,  250k income, no dependents, no spouse. They contributed 23,000 to 401k, 4,150 to HSA, 3,200 to FSA, and 5,000 to a backdoor Roth IRA. They received a 50% employer match up to 50% on 401k. ... If you have family HSA coverage and a dependent-care FSA, the total available would be 8,900.)  [boxed{2{,}200}]
>
> **Detailed Judge Requirements for the Response (Generated by Evo-Verify):**
> *(A full example will be included in the appendix; this is a conceptual summary)*
>
> > The verifier must perform a multi-step check:
> >
> > 1.  **401k Limit Check:** Verify that the model correctly identified the 2024 employee contribution limit ($23,000) and correctly determined that the user (age 49) is not eligible for the catch-up contribution.
> > 2.  **IRA Limit Check:** Verify the 2024 IRA limit ($7,000) and that the user's income ($250,000) makes them ineligible for a direct Roth contribution, validating the backdoor path. Check the calculation of remaining room.
> > 3.  **HSA Limit Check:** Verify the self-only HSA limit for 2024 ($4,150). The response's stated limit of $4,850 is an error (it's the 2025 limit), which the verifier must catch.
> > 4.  **FSA Limit Check:** Verify the 2024 Health FSA limit ($3,200) and note the response correctly identifies this is over-contributed.
> > 5.  **Final Calculation:** Verify that the final sum ($2,200) is correctly derived *from the response's own (flawed) premises*, but also note the factual error in the HSA limit.
> > 6.  **Output Format:** The verifier must output a JSON object containing a "correctness\_score" (float), a detailed "error\_analysis" string, and a "final\_verdict" (pass/fail).

---

### Note · Program_Chairs · 2026-01-17
**Submission Desk Rejected by Program Chairs**

The following references in this submission do not refer to real documents and/or have major errors in bibliographic information:

 Anonymous Authors. Reinforcement learning from human feedback: A survey. arXiv preprint arXiv:2401.06080, 2024e.
Leo Gao, John Schulman, and Jacob Hilton. Reward hacking in reinforcement learning from human feedback. arXiv preprint arXiv:2210.10760, 2023.
Anonymous Authors. Contrastive rewards for mitigating imperfections in reward models. arXiv preprint arXiv:2403.07708, 2024c.
Dae-Young-Kim, Seung-Hoon-Na, and Ju-Hyun-Lee. KorBench: A Comprehensive Benchmark for Korean Language Models, 2024.
Anonymous Authors. Challenges with human preference data in RLHF. OpenReview, 2024b. Based on various sources discussing data quality and bias.
Guanzheng Cui, Ziyi Liao, Yuanchun Wang, Ge Zhang, and Zhaofeng He. CompassArena: A Testbed for Political Typology Analysis and Bias Evaluation of Large Language Models, 2024.